# Gastrointestinal Incretins—Glucose-Dependent Insulinotropic Polypeptide (GIP) and Glucagon-like Peptide-1 (GLP-1) beyond Pleiotropic Physiological Effects Are Involved in Pathophysiology of Atherosclerosis and Coronary Artery Disease—State of the Art

**DOI:** 10.3390/biology11020288

**Published:** 2022-02-11

**Authors:** Szymon Jonik, Michał Marchel, Marcin Grabowski, Grzegorz Opolski, Tomasz Mazurek

**Affiliations:** Department of Cardiology, Medical University of Warsaw, Banacha 1a Str., 01-267 Warsaw, Poland; michal.marchel@gmail.com (M.M.); marcin.grabowski@wum.edu.pl (M.G.); grzegorz.opolski@wum.edu.pl (G.O.); tmazurek@kardia.edu.pl (T.M.)

**Keywords:** atherosclerosis, coronary artery disease, glucose-dependent insulinotropic polypeptide, glucagon-like peptide-1, dipeptidyl peptidase-4

## Abstract

**Simple Summary:**

The presented manuscript contains the most current and extensive summary of the role of the most predominant gastrointestinal hormones—GIP and GLP-1 in the pathophysiology of atherosclerosis and coronary artery disease both in animals and humans. We have described GIP and GLP-1 as (1) expressed in many human tissues, (2) emphasized relationship between GIP and GLP-1 and inflammation, (3) highlighted importance of GIP and GLP-1-dependent pathways in atherosclerosis and coronary artery disease and (4) proved that GIP and GLP-1 could be used as markers of incidence, clinical course and recurrence of coronary artery disease, and related to extent and severity of atherosclerosis and myocardial ischemia. Our initial review may state a cornerstone for the future, however, there are still many unknowns and understatements on this topic. Due to the widespread growing interest for the potential use of incretins in cardiovascular diseases, we think that further research in this direction is desirable. For the future, we would like to recognize GIP and GLP-1 as widely implemented into clinical practice as new biomarkers of atherosclerosis and coronary artery disease.

**Abstract:**

Coronary artery disease (CAD), which is the manifestation of atherosclerosis in coronary arteries, is the most common single cause of death and is responsible for disabilities of millions of people worldwide. Despite numerous dedicated clinical studies and an enormous effort to develop diagnostic and therapeutic methods, coronary atherosclerosis remains one of the most serious medical problems of the modern world. Hence, new markers are still being sought to identify and manage CAD optimally. Trying to face this problem, we have raised the question of the most predominant gastrointestinal hormones; glucose-dependent insulinotropic polypeptide (GIP) and glucagon-like peptide-1 (GLP-1), mainly involved in carbohydrates disorders, could be also used as new markers of incidence, clinical course, and recurrence of CAD and are related to extent and severity of atherosclerosis and myocardial ischemia. We describe GIP and GLP-1 as expressed in many animal and human tissues, known to be connected to inflammation and related to enormous noncardiac and cardiovascular (CV) diseases. In animals, GIP and GLP-1 improve endothelial function and lead to reduced atherosclerotic plaque macrophage infiltration and stabilize atherosclerotic lesions by directly blocking monocyte migration. Moreover, in humans, GIPR activation induces the pro-atherosclerotic factors ET-1 (endothelin-1) and OPN (osteopontin) but also has anti-atherosclerotic effects through secretion of NO (nitric oxide). Furthermore, four large clinical trials showed a significant reduction in composite of CV death, MI, and stroke in long-term follow-up using GLP-1 analogs for DM 2 patients: liraglutide in LEADER, semaglutide in SUSTAIN-6, dulaglutide in REWIND and albiglutide in HARMONY. However, very little is known about GIP metabolism in the acute phase of myocardial ischemia or for stable patients with CAD, which constitutes a direction for future research. This review aims to comprehensively discuss the impact of GIP and GLP-1 on atherosclerosis and CAD and its potential therapeutic implications.

## 1. Introduction

### 1.1. Atherosclrosis and Coronary Artery Disease

Acute coronary syndrome (ACS) and chronic coronary syndrome (CCS) are a group of clinical diagnosis related to myocardial ischemia and are both the manifestation of atherosclerosis in coronary arteries, named coronary artery disease (CAD). Atherosclerosis is the accumulation of cholesterol and fatty deposits (called plaques) at the inner walls of the coronary arteries. Atherosclerotic plaques are the cause of narrowing or complete blockage of the vessel lumen leading to decreased supply of oxygen-rich blood to the part of the heart muscle, causing its ischemia and necrosis. Additionally, atherosclerotic plaques may rapture, resulting in partial or complete thrombotic artery occlusion. The frequency of atherosclerotic plaque rupture is related to their size and lipid content. Atherosclerotic plaques, which narrow the vessel slightly to moderately, are more prone to rupture [1], leading to unstable angina (UA) or acute myocardial infarction (MI). Furthermore, increased lipid content is positively correlated with the incidence of plaque rupture, especially for those plaques localized eccentrically within the intima [1].

It is also known that atherosclerosis is a form of inflammation—atherosclerotic lesions are responsible for recruitment and production of cytokines, macrophages, neutrophils, lymphocytes, mast cells and many others [2]. Atherosclerosis is mainly a sequence of dyslipidemia that induces vascular cell dysfunction and thus causes inflammation, while the inflammatory process itself (without the accompanying dyslipidemia) may generate atherosclerotic plaque formation [2]. Hence, both lipid disorders as well as inflammation and pro-inflammatory molecules are potential therapeutic targets in atherosclerosis [2].

While ACS (consisting of such clinical conditions such as ST-elevation myocardial infarction (STEMI), non-ST-elevation myocardial infarction (NSTEMI), and unstable angina (UA)) are usually severely manifested, CCS may remain clinically and biochemically asymptomatic for a long time. If angina appears—it usually presents in the form of retrosternal pain, which is caused by physical effort or emotional stress—it disappears at rest or after taking nitrates; it is often severe in the morning and can be aggravated by cold air or plentiful meal, but it does not change depending respiratory phase or body position [3]. Crushing chest pain, often radiating to the left shoulder or angle of the jaw, sweating, nausea and vomiting, and shortness of breath are the first signs of acute occlusion of the culprit artery [4]. If such a condition, named MI, happens, the pain in the chest usually lasts more than 20 min and increases and does not disappear at rest or after taking nitrates. In some female patients, the elderly, and those with diabetes, symptoms may be non-specific and include anxiety, subfebrile condition, intercostal pressure, or abdominal pain [4].

Regardless of symptoms and clinical course, CAD is the most common cause of death in over 30% of people older than 35 years [5] and generally was responsible for 9.14 million deaths and affected 197 million of people in 2019 [6,7]. Although age-standardized prevalence and morbidity from CAD per 100,000 persons decreased between 1980 and 2019, especially in developed countries [8,9], it still remains the most serious medical problem of the modern world. Despite many clinical studies dedicated to it and an enormous development of diagnostic and therapeutic methods, new markers are still being sought to identify CAD easily and objectively for the implementation of optimal management as soon as possible.

### 1.2. Physiology of GIP and GLP-1

Glucose-dependent insulinotropic polypeptide (GIP) and glucagon-like peptide-1 (GLP-1) belong to a group of gastrointestinal hormones called incretins. Insulin released from the beta cells of the islets of Langerhans after ingestion of food is the major regulator of GIP and GLP-1 secretion [10]. Both hormones show complementary action of the β-cells of the pancreas by acting through different but related receptors [11]. GLP-1 is secreted from endocrine L-cells mainly found in the distal ileum and colon, while GIP is released from K-cells located in the duodenum and jejunum [11]. GLP-1 secretion is stimulated by activation of a number of intracellular signals, including PKA (protein kinase A), PKC (protein kinase C), calcium, and MAPK (mitogen-activated protein kinase) [11]. GLP-1 release is regulated by cell membrane channels: glucose stimulates GLP-1 secretion via K_ATP_ (adenosine triphosphate-sensitive potassium) channel closure [12], while nonmetabolizable carbohydrates using mechanisms dependent on sodium-glucose cotransporters 1 and 3 [13]. Many forms of GLP-1 are released in vivo, including GLP-1(1-37) and GLP-1(1–36)NH_2_, which are inactive, and GLP-1(7–37) and GLP-1(7–36)NH_2_, which are biologically active. GLP-1(7–36)NH_2_ consists of the majority of GLP-1 circulating in the human body [14,15].

The human GIP gene comprises six exons and has been localized on the long arm of chromosome 17 [11]. Previous studies demonstrated that bioactive form of GIP (42-amino acid) is released from its 153-amino acid proGIP precursor via PC 1/3 (prohormone convertase 1/3)-dependent posttranslational cut [16]. The GIP sequence is highly conserved, with more than 90% amino-acid sequence identity in humans and other vertebrates. [11] The GIP receptor (GIPR) initially was isolated from a rat brain, but nowadays we know that it is also expressed throughout the digestive tract, pituitary, lungs, trachea, kidneys, bones, thymus, spleen, endothelial cells, thyroid, adrenal cortex, testis, and central nervous system [17]. It was previously demonstrated that the secretion of GIP may increase via β-adrenergic stimulation, K^+^-mediated depolarization, rise of intracellular Ca^2+^ concentration, and activation of adenyl cyclase [18]. Studies have shown that ahead of gastrin, secretin, and cholecystokinin, GIP and GLP-1 are predominant incretin hormones in humans [10].

For the first time, the intestinal peptide GIP was isolated from porcine upper small intestine about 50 years ago [19], and it was discovered about 10 years later that two GLP-1 fragments were potent regulators of insulin secretion [20]. Both hormones are elevated after an oral glucose load or meal consumption in healthy humans and nowadays are commonly known for regulating glycemic level by increasing insulin secretion, inhibiting glucagon release, and delaying stomach emptying [21]. However, the increase in incretins after meals is disproportionately low in people with type 2 diabetes (DM 2) [22]. Although the reasons for this phenomenon are not fully explained, several mechanisms are postulated. Because insulin increase appears to be the main regulator of GIP and GLP-1 release, and insulin secretion in DM 2 patients is reduced due to beta cells dysfunction, it can be assumed that diminished effect of incretins in DM 2 can be associated with β-cell dysfunction rather than with a direct impairment of incretins secretion or activity [23,24]. In addition, acute elevation of glucose concentration (due to continuous insulin deficiency) in diabetic patients has been shown to reduce postprandial GLP-1 response [24,25]. Another postulated mechanism is specific loss of GIP activity. It may seem that glucose-induced insulin release impairment appears to be comparable to the corresponding GIP-induced insulin secretion defect. It is possible that the inability of GIP to increase insulin secretion during hyperglycemia is primarily due to the lack of glucose amplification of insulin release in DM 2 patients [26,27]. The insulinotropic GIP and GLP-1 inefficiency in diabetics is perhaps related with an additional (not yet explored) mechanism of action rather than with a specific defect in GIP regulation. Further research is required to clarify these mechanisms.

### 1.3. GIP and GLP-1—Importance in Medicine

For many years it was known that intravenous GIP or GLP-1 administration could normalize hyperglycemia in DM 2 patients [22]. Hence, GLP-1 receptor agonists (GLP-1 RAs) have been widely used as an oral treatment of DM 2 for several years [28,29]. Exenatide was introduced for the treatment of DM 2 in 2005 and liraglutide was approved for use four years later. [30] Besides these two, many other GLP-1 agonists like Semaglutide, Efpeglenatide, Dulaglutide, Albiglutide, and Lixisenatide have been recently synthesized [31]. Besides efficacy in reducing hyperglycemia, GLP-1 RAs have many additional benefits. One of their advantages over older insulin secretagogues, such as sulfonylureas or meglitinides, is lower risk of hypoglycemia development [32]. In addition, constant intake of GLP-1 stimulates weight loss and reduces appetite by inhibiting gastric emptying [32].

Apart from the undoubted impact of GIP and GLP-1 on carbohydrates metabolism, their receptors were also expressed in organs and cells such as duodenum, liver, kidneys, peripheral and central nervous system, adipocytes, osteoblasts, and myocytes [11,33].

#### 1.3.1. Liver

In the liver, both GIP and GLP-1 attenuate glucagon-stimulated glucose production [34]. Furthermore, it was discovered that glucose-induced GLP-1 secretion is decreased in patients with non-alcoholic fatty liver disease (NAFLD) and non-alcoholic steatohepatitis (NASH) [35].

#### 1.3.2. Kidneys

The GLP-1 receptor (GLP-1R) is found expressed in kidneys and intravenous administration of GLP-1 in rats has both a diuretic and a natriuretic effect with increases in glomerular filtration rate and inhibition of sodium reabsorption in the proximal tubule. Moreover, by enhancing the excretion of water and salt, GLP-1 demonstrates antihypertensive action [36]. Additionally, intravenous infusion of GLP-1 reduces H^+^ (hydrogen) secretion and glomerular hyperfiltration and thus protects the kidneys [37].

#### 1.3.3. Nervous System

In the central nervous system, GLP-1 inhibits the intake of food and fluids [38], decreases energy intake, and leads to weight loss by constant activation of satiety region in the hypothalamus [39]. Moreover, the anorectic effects of GLP-1 RAs have been described [40]. GLP-1R was also proved to be involved in many pathways for learning and memory abilities [41]. GLP-1 RAs have neuroprotective and anti-apoptotic effects on neuronal cells in a rodent model [42], and for this reason, GLP-1R can be proposed as a therapeutic target in many neurological disorders and neurodegenerative diseases. In the central nervous system, GIP is expressed mainly in the hippocampus; it has been suggested that administration of GIP induces proliferation of hippocampal progenitor cells and thus may enhance memory abilities and behavioral changes [43].

#### 1.3.4. Adipose Tissue

Incretins may display both lipogenic and lipolytic effects in human adipocytes. The anabolic effects of GIP include promoting fatty acid synthesis, insulin-dependent incorporation of fatty acids into triglycerides, enhancement of lipoprotein lipase production, and reduction of glucagon-stimulated lipolysis [44,45]. However, GIP may also act lipolitically by improving glucose tolerance, enhancing insulin sensitivity and reducing obesity-related pancreatic β-cells hyperplasia [46].

#### 1.3.5. Bones and Muscles

For the skeletal system, GIP is involved in bone formation and increases bone mineral density in rats [47]. These effects are mostly caused by increasing intracellular calcium (Ca^2+^) concentration. Additionally, GIP administration was found to be an inhibitor of bone resorption by retardation of osteoclasts proliferation [48]. In rat and human muscles, GLP-1 increases glycogen synthase activity and interferes with glucose metabolism [49].

#### 1.3.6. Endocrine System

Finally, incretins are also involved in the secretory activity of endocrine glands. GLP-1 stimulates a release of thyroid-stimulating hormone (TSH), luteinizing hormone (LH), corticosterone and vasopressin, and modulates neuroendocrine cells in hypothalamic rats [50]. Although it has not been definitely confirmed that GIP regulates cortisol release in healthy subjects, abnormal expression of GIPR was found to be related with development of Cushing’s syndrome [51].

#### 1.3.7. Inflammation

Moreover, studies describing the beneficial anti-inflammatory effect of GIP and GLP-1 can be found in the literature [52]. GIPR and GLP-1R are widespread expressed on many immune cells, and it was proved that GIP and GLP-1 can activate the immune system to restrain disease processes [53]. On the other hand, GLP-1 concentrations were found to be higher in critically ill patients with sepsis and positively correlated with proinflammatory markers [54].

### 1.4. Dipeptidyl Peptidase-4 (DPP-4) Role

Both GIP and GLP-1 are rapidly inactivated by the enzyme dipeptidyl peptidase-4 (DPP-4) with a blood half-life of only a few minutes [22]. DPP-4, also known as CD 26 (cluster of differentiation 26), was discovered over 50 years ago [55]. DPP-4 belongs to the adipokines family and appears an important molecular biomarker that is strongly correlated with metabolic syndrome [56]. DPP-4 expression was proved to positively correlate with body mass index (BMI) and adipocyte size and activity in both subcutaneous adipose tissue (SAT) and visceral adipose tissue (VAT) [56]. In the same study, circulating DPP-4 concentration in insulin-sensitive obese patients was significantly lower than in those with insulin resistance [56]. However, it has not yet been fully explained which tissue is the source of circulating DPP-4. It is also known that DPP-4 is involved in metabolism and release of broad range of molecules such as chemokines, vasoactive peptides, neurokinins, and growth factors [57]. DPP-4 is also associated with cancer biology and is useful as a marker of various tumors, with its level strongly correlated with neoplasia [58]. Additionally, DPP-4 inhibitors were found to have an anti-inflammatory effect via inhibition of monocyte activation and chemotaxis [59] and are involved into inhibition of the development of endothelial dysfunction and thus prevent atherogenesis in nondiabetic apolipoprotein E-deficient (ApoE^−/−^) mice [60].

Apart from pleiotropic actions, DPP-4 plays the most important role in glucose metabolism [61]. By breaking down endogenous incretins GIP and GLP-1, it increases plasma glucose level both in fasting and fed conditions and is also responsible for less postprandial insulin burst [62]. Therefore, drugs that are DPP-4 inhibitors delay the breakdown of endogenous incretins GIP and GLP-1 and are successfully used to combat insulin-resistance and DM 2 management [29].

GIP, GLP-1 and DPP-4 inhibitors have not only positive pleiotropic effects on the whole human body. Incretin-based therapy has recently been shown to be associated with an increased risk of pancreatic cancer in diabetes patients [63].

The currently known physiological effects of GIP, GLP-1, and DPP-4 on animal and human tissues are presented on Figure 1.

The recently published results of the CAPTURE study provide real-life evidence for importance of incretins in medicine, demonstrating that the vast majority of DM 2 has an atherosclerotic form of cardiovascular disease (CVD). In this multicenter trial assessing worldwide prevalence of CVD in nearly 10,000 people with DM 2, CVD was diagnosed in one in three patients with DM 2, whereas overall prevalence rates of weighted and atherosclerotic CVD were 34.8% and 31.8%, respectively. Furthermore, in this trial, GLP-1RA or SGLT-2 inhibitors were proved to play beneficial CV effects even in patients with established CVD [64].

## 2. Current Knowledge in Atherosclerosis and Coronary Artery Disease Insights from Animal Studies

### 2.1. Atherosclerosis

Exendin-4 (one of the firstly synthetized GLP-1 RA) is involved in the accumulation of monocytes and macrophages into the arterial wall, which represents an early phase of atherosclerosis. This effect was described by Arakawa et al. in 2010 [65]. After administration of low or high doses of exendin-4 in C57BL/6 (control) or ApoE^−/−^ mice, the monocyte adhesion to the endothelia of thoracic aorta and arteriosclerotic lesions around the aortic valve were evaluated. In this study, Arakawa et al. provide evidence that GLP-1 agonism prevents the progression of atherosclerosis in ApoE^−/−^ mice by reducing the expression of inflammatory mediators TNF-α (tumor necrosis factor α) and MCP-1 (important cytokines and chemokines with established atherogenic effect) in activated macrophages without major effects on metabolic parameters. Furthermore, by using adenylate cyclase inhibitor and activator, researchers demonstrated in this study that the stimulation of cAMP (cyclic adenosine monophosphate) by exendin-4 is critical for the attenuated production of proinflammatory mediators from macrophages. These data suggest that exendin-4 regulates inflammatory response of macrophages via the cAMP/PKA pathway, which inhibits proinflammatory cytokine production.

In a study by Nagashima et al. [66] investigating the potential possibility of gastrointestinal hormones against protection from atherosclerosis, active forms of incretins—GLP-1(7–36)amide and GIP(1–42)amide were proved to exert an anti-atherogenic effect. A total number of 346 apolipoprotein E knockout (ApoE^−/−^; ApoE^−/−^ mice under normal circumstances spontaneously develop atherosclerosis) male mice from 17 weeks of age were divided into nine groups and were started to be, respectively, infused for 4 weeks with active or inactive forms of GLP-1 and GIP or saline (vehicle). Administration of GLP-1(7–36)amide and GIP(1–42)amide significantly suppressed atherosclerotic lesions and macrophage infiltration in the aortic wall compared with vehicle controls. The molecular mechanisms underlying these effects were described as linked with significant decreases in foam cell formation followed by cAMP activation and the downregulation of CD36 (cluster of differentiation) and ACAT-1 (acyl-coenzyme A:cholesterol acyltransferase-1) by active incretins. The inactive forms of GLP-1 and GIP had no effects on atherosclerosis and macrophage foam cell formation.

In 2012, Nogi et al. [67] demonstrated a significant anti-atherogenic effect of GIP in diabetic animals. Nondiabetic ApoE^−/−^ mice, streptozotocin-induced diabetic ApoE^−/−^ mice, and db/db (a mouse model of type 2 diabetes) mice were administered GIP or saline (vehicle) through osmotic mini-pumps for 4 weeks. Diabetic ApoE^−/−^ mice exhibited more advanced atherosclerosis than a nondiabetic ApoE^−/−^ group of the same age. Importantly, GIP infusion remarkably reduced the surface areas of the atherosclerotic lesions and suppressed the atheromatous plaque size and macrophage infiltration in the aortic root, as compared with vehicle-infused counterparts. Foam cell formation in macrophages from GIP-infused diabetic ApoE^−/−^ mice was also suppressed significantly in comparison with that in the vehicle-infused control. It should be highlighted that both these GIP-induced effects were significantly abolished by the co-infusion with [Pro^3^]GIP, a GIPR antagonist.

A similar effect of incretins was demonstrated by Terasaki et al. [68] who attempted to study the association between the DPP-4 inhibitor and atherosclerosis in a murine model, but animals were administrated with DPP-4 inhibitor orally. Seventeen-week-old ApoE^−/−^ mice were fed with an atherogenic diet and were supplied a DPP-4 inhibitor, vildagliptin analogue (PKF275-055; PKF) in drinking water over a period of 4 weeks. Compared with drinking water controls, PKF significantly suppressed total aortic atherosclerotic lesions, atheromatous plaque in the aortic root, and macrophage accumulation in the aortic wall, but none of these results were linked with the PKF-induced reductions in body weight and plasma cholesterol levels.

Another study evaluating functional relevance of GLP-1 and atherosclerosis in murine model was conducted by Burgmaier et al. [69]. Various forms of GLP-1 (GLP-1(7–37), GLP-1(9–37) and GLP-1(28–37)) and LacZ (control) were overexpressed for a period of 12 weeks in ApoE^−/−^ mice on a high-fat diet (*n* = 10 per group) using an AAV vector system. All groups with GLP-1 overexpression were found to present with significantly reduced plaque macrophage infiltration and plaque MMP-9 expression as compared to LacZ in the aortic arch and root; however, neither any of GLP-1 forms or LacZ changed overall atherosclerotic lesion size. Moreover, all GLP-1 constructs increased plaque collagen content and increased fibrous cap thickness, which can be translated into improved plaque stability.

Furthermore, to explore the functional connection of elevated GIP in human atherosclerotic lesions, Kahles et al. overexpressed GIP(1–42) in ApoE^−/−^ mice by injecting adeno-associated viral (AAV) vector system via tail veins. [70] Mice were switched to a Western diet 4 weeks after intravenous vector injection, which they remained on for a total of 12 weeks. Compared to control (LacZ; vectors carried transgene cassettes encoding b-galactosidase), viral overexpression of GIP(1–42) resulted in a significant elevation of serum GIP at day 21 after injection without affecting body weight, glucose tolerance, and serum lipids. GIP overexpression reduced atherosclerotic plaque macrophage infiltration and increased collagen content but did not affect total atherosclerotic lesions size and lipid content per plaque in the descending aorta and the aortic root and arch. Furthermore, treatment with GIP was demonstrated to reduce monocyte chemotactic protein-1 (MCP-1)-induced monocyte migration, lipopolysaccharides, endotoxin (LPS)-induced interleukin-6 (IL-6) secretion, and matrix metallopeptidase-9 (MMP-9) activity. All these mechanisms were found to be responsible for attenuating atherosclerotic plaque inflammation or instability in mice model.

Finally, it should be highlighted that GIP’s role in peripheral vasculogenesis was also studied (Mori Y) [71]. In mice, treatment with GIP(1–42) suppressed neointimal hyperplasia as compared with vehicle without affecting medial area or arterial perimeter, resulting in a reduction in the ratio of neointimal area to medial area. The investigators suggested nitric oxide (NO) as responsible for this effect, providing that neointimal hyperplasia and cell proliferation were greater in L-NAME (inhibitor of NOS (NO synthase))—treated mice than in vehicle—treated cohort. Moreover, it was described that the activation of AMPK (5’AMP-activated protein kinase) is involved in GIP-stimulated NO production and CaMKK (calcium-calmodulin–dependent protein kinase kinase) and PLC (phospholipase C) mediate the activation of AMPK by GIP.

### 2.2. Myocardial Ischemia

In the very first research directly regarding this topic, Bose et al. [72] investigated the effects of GLP-1 infusion in vitro in rats subjected to 30 min of left main (LM) artery occlusion and 2 h of reperfusion and observed that GLP-1 had no salutary effects on hemodynamics. The absence of hemodynamic benefits was surprising given the fact that the investigators found a 50% reduction in infarct size in the GLP-1-treated cohort as compared with the valine pyrrolidide (VP; DPP-4 inhibitor) or saline groups. For the first time, GLP-1 was demonstrated to protect against myocardial infarction in the isolated and intact rat heart. Additionally, investigators showed that the cAMP inhibitor Rp-cAMP abolished protection, confirming this known GLP-1 pathway as a possible mechanism. The protection was also suppressed by the PI3K (PhosphoInositide 3-Kinase) inhibitor LY294002 and by the p44/42 mitogen-activated protein kinase inhibitor UO126, implicating both these well-known prosurvival proteins in the cardioprotection mediated by GLP-1. Each of these mechanisms appears to be essential for the preservation afforded by GLP-1, as inhibiting them individually abrogates the protection, suggesting that they may act in parallel.

Further, about 15 years ago, the clinical relevance between GLP-1 and myocardial ischemia was described by Zhao et al. in rats [73]. LV (left ventricular) function and myocardial glucose uptake were assessed in 75 male rats under basal conditions and after 30 min of low-flow ischemia and 30 min of reperfusion in the presence and absence of GLP-1(7–36)amide. In normal, non-ischemic hearts, GLP-1 decreased LV developed pressure (LVdp) and increased coronary flow but did not change LV end-diastolic pressure (LVEDp) or heart rate. GLP-1 treatment intensified recovery after 30 min of low-flow ischemia, leading to a faster increase in LVdp and lower postischemic LVEDp. Furthermore, in non-ischemic conditions, GLP-1 significantly increased myocardial glucose uptake by increasing nitric oxide (NO) production and glucose transporter (GLUT)-1 translocation to a greater extent than in the control or insulin-treated groups. However, during reperfusion, both GLP-1 and insulin increased LV function, myocardial glucose uptake, and GLUT-1 and GLUT-4 translocation similarly.

The GLP-1 and its association with experimental myocardial infarction in porcine model was also evaluated by Timmers et al. in 2009 [74]. In this study, 18 pigs were randomly assigned to treatment with exenatide (GLP-1 RA) or phosphate-buffered saline (PBS) after 75 min of myocardial ischemia by left circumflex artery (LCx) ligation and subsequent reperfusion for 3 days. An exenatide administration before reperfusion significantly reduced myocardial infarct size by 40% compared with the PBS group. Furthermore, the end-diastolic (ED) and end-systolic (ES) volumes were lower in exenatide-treated pigs as compared with controls, indicating that exenatide prevented acute LV dilation after MI and maintained contractile performance. Additionally, other global functional parameters, such as LV ejection fraction (LVEF), dP/dt max (mmHg/s), and ES elastance were higher after exenatide treatment. In addition, myocardial stiffness was lower, indicating that diastolic function also improved. What is more, the ED pressure increased significantly in PBS-treated animals but not in pigs treated with exenatide. Finally, after exenatide treatment, myocardial phosphorylated Akt (pAkt; a kinase linked to insulin signaling pathway) and antiapoptotic B-cell lymphoma 2 (Bcl-2) expression levels were higher, and activity of superoxide dismutase and catalase (antioxidant enzymes) were increased compared with those after PBS treatment, while active caspase 3 (inductor of apoptosis) expression was lower.

By exploring molecular mechanisms on this issue, Lee et al. showed that GIP, by attenuating some mediators involved in insulin resistance, may act cardioprotectively [75]. In this study, normoglycemic male rats after LAD ligation were randomized to either vehicle or sitagliptin (DPP-4 inhibitor) for 4 weeks starting 24 h after operation. The investigators demonstrated that myocardial infarction was associated with an increased resistin (an adipokine known to be correlated with inflammation and atherosclerosis) expression via the GIP-dependent pathway. They found that infusion of infarcted hearts with GIP resulted in similarly decreased levels of resistin as compared with sitagliptin, whereas GLP-1 did not have such an effect, suggesting that the metabolism of resistin was GLP-1-independent. Moreover, sitagliptin by DPP-4 inhibition reduced adverse effects of resistin by Akt protein/PI3K signaling, which is a GIP-dependent pathway.

Additionally, in animals (experimental murine model of MI by left anterior descending (LAD) ligation), Ussher et al. (2017) demonstrated that GIP is involved in myocardial injury via some pathways regulating cardiac lipid metabolism [76]. The cardiomyocyte GIPR regulates fatty acid metabolism (direct antagonism of GIPR decreased fatty acid oxidation) and the adaptive response to ischemic cardiac injury. A genetic elimination of the GIPR (GIPR^−/−^ mice) reduced ventricular injury and adverse remodeling following experimental MI and enhanced survival associated with reduced hormone sensitive lipase (HSL) phosphorylation; it also increased myocardial triacylglycerol (TAG) stores. Overexpression of HSL reversed the cardio-protective effect of GIPR antagonism, suggesting HSL suppression and increased triglycerides as the mechanism by which GIPR antagonism protects from ischemic-induced mortality. However, the mechanism of action behind the potential cardioprotective effect of triglycerides remains unknown.

Finally, in a study by Diebold et al., circulating concentrations of GLP-1 were assessed after experimental MI and were evaluated in light of metabolism, LV contractility, and mitochondrial function [77]. The investigators found that permanent LAD ligation in the murine model enhanced total GLP-1 serum levels in a time-dependent manner, with a maximal rise 6 h post-procedure, while no difference in circulating glucose or insulin concentrations between LAD-ligated and sham-operated animals was observed. Furthermore, GLP-1 significantly increased LV contractility as early as the 3 h time point, which remained significant 6 h post permanent LAD ligation. Simultaneously, pre-treating with exendin-9 (GLP-1R antagonist) impaired LV contractility 6 h post LAD ligation and abrogated the beneficial effect of linagliptin-dependent DPP-4 inhibition without affecting the heart rate. Furthermore, DPP-4 inhibition increased AMPK activity and stimulated the mitochondrial respiratory capacity of non-infarcted tissue areas. Thus, we can conclude that GLP-1 increased mitochondrial respiration in non-infarcted tissue, which may provide the energetic capacity to augment LV contractility during MI.

The summary of studies involving effects of GIP and GLP-1 on atherosclerosis and myocardial ischemia in animal models is presented in Table 1.

## 3. Current Knowledge in Atherosclerosis and Coronary Artery Disease—Insights from Human Studies

### 3.1. Atherosclerosis

The first randomized-controlled study investigating the association between GLP-1 RA and oxidative stress was by Bunck et al. Sixty DM 2 patients treated with metformin were randomized to receive exenatide or glargine insulin (ratio 1:1) for one year. A 51-week exenatide treatment resulted in significant reduction of plasma concentration of oxidative stress markers: malondialdehyde–MDA (compared to baseline and glargine insulin; *p* < 0.001) and oxidized low-density lipoprotein–oxLDL (compared to baseline; *p* < 0.05). Moreover, the investigators reported a significant decrease in postprandial glucose and triglycerides in the exenatide cohort as compared to baseline and glargine insulin (*p* < 0.01). Changes in markers of oxidative stress were independent of the treatment arm and were related to changes in postprandial glucose and triglyceride excursions [78].

In 2014, Rizzo et al. presented the effect of GLP-1 RA–liraglutide on subclinical atherosclerosis. In this prospective study, 64 DM 2 patients with no prior history of CAD were evaluated. The investigators aimed to determine if adding liraglutide to metformin affects carotid-intima media thickness (CIMT). After 8 months with liraglutide treatment, CIMT significantly decreased from 1.19 to 0.94 (*p* = 0.001), while no significant reductions in body weight or waist circumference were observed. However, adding liraglutide to metformin resulted in significant decrease in fasting glucose (−2.1 mmol/L, *p* < 0.01) and HbA1C (−1.9%, *p* < 0.01). These results should, however, be considered carefully as there was no placebo group [79]. Furthermore, Zhang et al., in the first randomized-controlled trial, demonstrated the beneficial effect of GLP-1 RA on subclinical atherosclerosis via preventing atherosclerotic progression, as assessed by CIMT. A total number of 66 DM 2 patients were randomized to receive twice-daily exenatide or aspartate insulin for 52 weeks. Exenatide was proved to reduce the CIMT from baseline more significantly compared with insulin after 52 weeks with a mean difference of −0.14 mm (*p* = 0.016). The components of an explanatory endpoint (differences in body weight and lipid metabolism markers) were also more significantly reduced by exenatide at weeks 16 and 40 as compared with insulin. The correlation analyses showed that CIMT was positively associated with low-density lipoprotein (LDL) cholesterol [80].

In the study by Ceriello et al. [81], both type 2 diabetic patients (*n* = 16) and healthy matched control subjects (*n* = 12) underwent the following studies: a standard meal and an oral glucose tolerance test (OGTT; 75 g glucose in 300 mL water) in randomized order on different days. These tests were followed in a randomized order and on different days by two hyperglycemic clamps with or without GLP-1. The clamp studies were then repeated randomly with the same levels of glycemia and GLP-1 infusion rate. As expected, basal glycemia, insulin, HbA1c (glycated hemoglobin), and markers of oxidative stress (nitrotyrosine and 8-iso-PGF2a (8-iso prostaglandin F2a)) were increased in diabetes, and FMD (flow-mediated vasodilation; a marker of endothelial function) was decreased. Basal, fasting level of GLP-1 was not different between control subjects and diabetic patients. During OGTT, an increase in glycemia, nitrotyrosine, and 8-iso-PGF2a and a decrease in FMD were observed in the control subjects and in diabetic patients. During the clamps, performed with a placebo, GLP-1 concentration remained unchanged during the study period in both control subjects and diabetic patients. However, in both control subjects and diabetic patients, the values of nitrotyrosine and 8-iso-PGF2a significantly increased, and the values of FMD significantly decreased in the clamp with placebo compared with the values observed for the clamp with GLP-1. In this way, Ceriello et al. demonstrated that the presence of GLP-1 during hyperglycemia significantly protects endothelial function and decreases hyperglycemia-induced oxidative stress generation. In the absence of GLP-1, hyperglycemia induces endothelial dysfunction and oxidative stress, whereas the concomitant infusion of GLP-1 significantly prevents this effect.

Furthermore, in a retrospective study by Berglund et al. [82], plasma from patients with confirmed diagnosis of critical limb ischemia (*n* = 85) and healthy controls (*n* = 101), as well as mouse aortas and carotid arteries, were analyzed for proatherogenic cytokine–OPN (osteopontin), ET-1 (endothelin-1) and GIP levels. Human carotid plaques and plasma were collected at carotid endarterectomies. Additionally, GIP levels were measured in plasma from patients with CV disease or DM 2 and healthy control subjects. The researchers demonstrated that GIP stimulated the expression of OPN in mouse arteries ex vivo by a mechanism involving the release of ET-1 and activation of CREB (cAMP response element binding protein). Furthermore, plasma ET-1 and OPN levels were positively correlated in patients with critical limb ischemia, and infusion of GIP increased plasma levels of OPN. Fasting GIP levels were significantly higher in patients with a history of CV disease (MI or stroke) when compared to controls, and patients with symptoms of cerebral ischemia exhibited higher plaque GIPR and OPN mRNA levels and higher plasma OPN than asymptomatic patients.

### 3.2. Coronary Artery Disease

For the first time in humans, Nikolaidis et al. [83] demonstrated that 72 h infusion of GLP-1 for acute MI with successful reperfusion increased LVEF and infarct-zone-related regional wall motion. This prospective, nonrandomized study included 21 patients (11 after successful primary coronary angioplasty and 10 controls). Baseline demographics and background therapy were similar, and both groups had severe LV dysfunction at baseline. Global LVEF improved from 29.2% to 39.2% (*p* < 0.01) in the GLP-1–treated patients but not in the control group (28.2% to 29.2%). Furthermore, global wall motion score index (g-WMSI) and regional wall motion indexes (r-WMSI) (by definition, lower scores are associated with better contractile function) were significantly decreased in GLP-1 group (*p* < 0.01) as compared with control subjects. The benefits of GLP-1 were independent of MI location or history of diabetes. However, this study was limited by small sample size and nonrandomized character.

A larger, randomized, double-blind, placebo-controlled trial with 172 patients (6.4% diabetic) presented within 12 h from the onset of symptoms and signs of STEMI [84]. All of them underwent primary coronary angioplasty. It was demonstrated that six-hours exenatide (GLP-1 RA) administration initiated 15 min before onset of reperfusion reduced the ischemic area. However, no differences in LVEF at 3 months and adverse 30-day clinical events were observed. In the intention-to-treat analysis, assessment of the secondary endpoint of peak troponin T levels showed no difference between the exenatide and placebo groups (6.7 + 5.4 vs. 6.2 + 5.4 mg/L; *p* = 0.50).

In another prospective, randomized trial, with 58 patients admitted to hospital for STEMI and with thrombolysis in myocardial infarction (TIMI) flow 0, it was reported that, besides infarction size and improved LVEF, the releases in creatine kinase-MB (CKMB) and troponin I (TnI) were also significantly reduced after exenatide application [85].

We have previously described the findings of Kahles et al. involving the effect of GIP(1–42) overexpression in ApoE^−/−^ mice [70]. In this study, the association between GIP levels and CAD and PAD (peripheral artery disease) were also assessed. Serum concentrations of GIP were measured in 731 patients who presented for elective coronary angiography. Kahles et al. demonstrated increased serum concentration of circulating GIP levels in patients with PAD as compared with controls (413.0 vs. 332.7 pg/mL; *p* = 0.0165), and no link between GIP and higher incidence of CAD was found. It should be highlighted that GIP levels were independently related to PAD after multivariable adjustment not only for diabetes, but also for CAD, age, sex, BMI, hypertension, and smoking.

Furthermore, several large, prospective, randomized clinical trials aimed to investigate the association between treatment with GLP-1 RAs and prognosis of DM 2 patients with recognized CV disease or high CV risk. A significant reduction of primary composite endpoint (CV death, nonfatal MI, and nonfatal stroke) in long-term follow-up using GLP-1 RAs for these patients were demonstrated for liraglutide in LEADER, semaglutide in SUSTAIN-6, dulaglutide in REWIND, and albiglutide in HARMONY [86,87,88,89], but not in PIONEER-6 (semaglutide) and EXSCEL (exenatide) and ELIXA (lixisenatide), in which GLP-1 RAs was only noninferior to placebo for primary composite endpoint: CV death, nonfatal MI and nonfatal stroke in PIONEER-6 [90] and EXSCEL [91], and composite of CV death, nonfatal MI, nonfatal stroke, and UA in ELIXA [92].

In a very recent study by Trevisan et al. from SWEDEHEART registry during 2010–2017 [93], the influence of GLP-1 RAs and the risk of CV events in diabetic patients surviving an acute MI were evaluated. Out of 17,868 discharged diabetes patients who were alive after a first event of MI, 365 (2%) were using GLP-1 RAs, which was associated with a lower event risk after median 3 years of follow-up (adjusted HR 0.72), mainly attributed to a lower risk of reinfarction and stroke with no suggestion of heterogeneity across subgroups of age, sex, chronic kidney disease, and STEMI.

The meta-analysis of Bethel et al. [94] evaluated CV outcomes for DM patients using GLP-1 RAs; four randomized trials (ELIXA, LEADER, SUSTAIN-6 and EXSCEL) were included. Compared with the placebo group, patients treated with GLP-1 RAs showed a significant reduction in primary composite endpoint (CV death, nonfatal MI and nonfatal stroke; HR 0.90, *p* = 0.033), CV death (HR = 0.87, *p* = 0.007), and overall mortality (HR = 0.88, *p* = 0.002), with no significant reduction of fatal and nonfatal MI, fatal and nonfatal stroke, hospitalizations for UA or HF.

Summarizing the above reports, it should not be ignored that for DM 2 patients with no established CV diseases but very high/high CV risk, GLP-1 RAs are also associated with CV risk reduction. In a recent trial-level meta-analysis, Marsico et al. [95] investigated the effects of GLP-1 RAs on major CV events in DM patients with and without established CV diseases. In the analysis of the whole population of DM patients (7 randomized controlled trials, 56,004 individuals), with no significant difference (HR = 1.06) in efficacy with respect to the major adverse cardiovascular events (MACE) primary endpoint (including CV mortality, non-fatal MI, and non-fatal stroke) between patients with established CVD and patients with CV risk factors only, GLP-1 RAs showed a significant 12% reduction in the hazard of the three-point MACE composite endpoint (HR = 0.88) and a significant reduction in the risk of CV mortality (HR = 0.88), all-cause mortality (HR = 0.89), fatal and nonfatal stroke (HR = 0.84), and HF hospitalization (HR = 0.92,) with no significant effect observed for fatal and nonfatal MI (HR = 0.91). Finally, the association between GLP-1 and lipid profile was also investigated. In the study by Anholm et al., the effect of liraglutide combined with metformin on lipid particle sub-fractions in 41 patients (28 with complete follow-up) with newly diagnosed DM 2 and at high risk of CV events, i.e., with CAD. Overall, liraglutide did not affect lipid subfractions or markers of low-grade inflammation (LGI) compared to placebo. However, the combination of liraglutide and metformin significantly reduced the most atherogenic subfraction–small, dense LDL_5_ particles despite the population being well-controlled on a stable statin therapy. Thus, the investigators proved that antidiabetic drugs using for DM 2 patients with CAD may reduce CV risk and improve lipid profile by lowering the level of small dense LDL [96].

### 3.3. Some Studies Have Additionally Focused on the Association between GLP-1 and Prognosis of Non-Diabetic Patients with ACS

In a study by Blatt et al. [97], blood plasma for GLP-1 levels was determined from 12 consecutive patients (mean age (SD) = 61.9 (12.0) years; 100% male; 10—non-diabetic) with STEMI before and 24, 72 h, and 90 days after primary PCI. The authors reported that mean group GLP-1 levels 24 h after arrival as well as peak levels determined within 72 h were significantly increased from 27 ± 7.1 to 39.5 ± 11.4 and 43.4 ± 11.1 pmol/L as compared with the mean levels before performance of PCI (*p* < 0.04 and *p* < 0.006, respectively). The mean GLP-1 levels in these 12 patients determined 90 days after discharge did not significantly differ as compared to levels at arrival. Only the history of hypertension and smoking were correlated with significantly lower levels of GLP-1 as compared with normotensive (60.1 ± 20.075 vs. 26.78 ± 5.75) and non-smokers (58.27 ± 17.13 and 22.68 ± 4.413 pmol/L, *p* < 0.01 and *p* < 0.04, respectively), but no correlation was found between GLP-1 levels and any of the clinical and laboratory parameters.

Recently, the research by Kahles et al. [98] demonstrated that GLP-1 is a powerful biomarker of CV events and death in patients with MI. In this study, 918 patients hospitalized for MI (321 with STEMI and 597 with NSTEMI; mean age (SD) = 66.9 (12.7) years; 73.2% male) were observed for CV outcomes in long-term follow up. It is worth emphasizing that 695 (75.7%) of the enrolled patients were non-diabetic. The primary composite endpoint of the first occurrence of non-fatal MI, non-fatal stroke or CV death was observed in 62 patients (7%), while all-cause mortality was observed in 68 patients (7%). Median follow-up was 310 days for the combined triple endpoint and 311 days for all-cause mortality. GLP-1 was found to be associated with combined primary endpoint and all-cause mortality (hazard ratio (HR) of logarithmized GLP-1 values: 6.29; *p* < 0.0001; HR: 5.71; *p* < 0.0001). GLP-1 was found to be a strong indicator of CV risk, especially for early events (30 days from admission), for which it was proved to be superior to other established biomarkers, including high sensitive Troponin T (hs-TnT), glomerular filtration rate (GFR), high sensitive C-reactive protein (hs-CRP), and N-terminal pro-B-type natriuretic peptide (NT-proBNP).

Another research study, presented earlier in animal studies, explored the role of endogenous GLP-1 in response to acute MI [77].

A total of 41 patients with clinical indication for coronary angiography (26-STEMI; 15-control (angiographic exclusion of CAD)) were assessed. Diebold et al. found endogenous circulating GLP-1 concentrations to be elevated in patients with STEMI, independently of food intake. Furthermore, activation of GLP-1 system was demonstrated as increasing LV contractility during MI.

In a study by Greener et al. [99], 103 patients with STEMI (*n* = 33; 20—non-diabetes) and three control groups: NSTEMI (*n* = 27; 14—non-diabetes), stable angina pectoris (*n* = 18; 8—non-diabetes), and healthy subjects (*n* = 25) as a control group were assessed. Plasma levels of total and active GLP-1 and soluble DPP-4 (sDPP4) were estimated by ELISA on admission and 24, 48 h after PCI in all patients. It was demonstrated that admission levels of GLP-1 were significantly increased in patients with STEMI, NSTEMI, and stable angina pectoris as compared with healthy individuals (*p* < 0.05). At 24 and 48 h, only STEMI and NSTEMI patients demonstrated a significant increase in active GLP-1 levels compared with admission levels (*p* < 0.05) and healthy controls (*p* < 0.0005). It is worth nothing that no correlations between the GLP-1 levels and any clinical laboratory and demographic parameters, including, age, gender, heart rate, number of atherosclerotic vessels, diabetes duration, glucose, HbA1c, and kidney function in the entire cohort were found. The authors noted that a small study group did not allow for adequate determination of correlation between GLP-1 and clinical outcomes; hence the effect of GLP-1 on prognosis of patients with STEMI or NSTEMI remains unknown. Additionally, the influence of glucose intolerance, which is commonly found in non-diabetic patients during acute phase of myocardial ischemia was not evaluated.

The summary of results from human studies evaluating the role of GIP and GLP-1 in atherosclerosis and coronary artery disease were presented in Table 2.

## 4. The Dual GIP/GLP-1 Agonism

Very recently, there have been reports in the literature of an effect of dual agonists (GIP/GLP-1) having a positive impact on glycemic control and improving insulin sensitivity, thus reducing complications related to hyperglycemia. In several animal studies, it was demonstrated that co-administration of GLP-1 and GIP improved hypoglycemic, insulinotropic, and total body-mass-reducing effects as compared with individual incretins [100,101]. Moreover, the synergistic action of both GLP-1 and GIP was proved to improve energy consumption and food intake in mice [100,101,102,103,104]. However, most of these studies did not evaluate the role of glucagon, mainly because its level in these studies was not measured. Human studies provide more data on this topic. The co-infusion of GIP/GLP-1 increased the insulin secretion compared with single use of GLP-1 or GIP in a study by Nauck et al. [105]. Moreover, the co-administration of incretins induced a significant glucagonostatic effect compared to separate infusion of glucose alone [106]. In another study, the insulin level increased after co-infusion as compared with GIP, GLP-1, or saline (placebo) alone in healthy subjects. However, in obese patients with DM 2, the insulin secretion did not rise as compared with GLP-1 infusion alone, but an increase when compared with GIP and placebo administration was detected, which highlighted the reduced insulinotropic effect of GIP in diabetic subjects [107]. In the study by Mentis et al., the co-infusion of GLP-1 and GIP as compared with GLP-1 alone resulted in similar blood glucose and insulin secretion rates in diabetic patients, while the suppression of plasma glucagon by GLP-1 was antagonized by GIP [108]. After that, since the co-infusion of GLP-1 and GIP was proved as having a favorable effect on glucose and bodyweight control, studies concerning the dual molecule GLP-1R/GIPR as a twincretin have been initiated. In 2013, Finan et al. demonstrated that unimolecular dual incretin administered to rodent models of obesity and diabetes improved insulin sensitivity and increased pancreatic insulin deficiency, corrected hyperglycemia and adiposity-induced insulin resistance, and lowered body weight [109]. The beneficial effect of GLP-1R/GIPR dual agonist was further confirmed in clinical trials. The double-blind, placebo-controlled trial with a fatty-acylated GIP/GLP-1 dual agonist (NNC0090-2746, Phase 1 and Phase 2) was performed in patients with DM 2 inadequately controlled with metformin administered with 1.8 mg of NNC 0090-2746 as compared with 1.8 mg of liraglutide. After two weeks of treatment, NNC0090-2746 decreased HbA1c, fasting, and postprandial plasma glucose. NNC0090-2746 reduced HbA1c similarly to the group treated with liraglutide, whereas the bodyweight reduction was significantly greater than that of liraglutide. The treatment with NNC0090-2746 was generally safe and well tolerated [110]. Furthermore, another GLP-1R/GIPR dual agonist, LY3298176, was also tested (Phase 2). This double-blind, randomized study was performed in DM 2 subjects, who received either once weekly subcutaneous LY3298176 (1 mg, 5 mg, 10 mg, or 15 mg), dulaglutide (1.5 mg), or placebo. After 26 weeks of treatment, LY3298176 significantly reduced HbA1c in a dose-dependent manner and reduced body weight as compared with dulaglutide or placebo with similar to dulaglutide adverse effects and no reports of severe hypoglycemia [111]. Based on promising results from Phase 1 and 2 clinical trials, a Phase 3 clinical trial of the dual GIP/GLP-1 RA (LY3298176) named tirzepatide was initiated. To date, the SURPASS program includes several ongoing randomized controlled trials for DM 2 patients: tirzepatide in monotherapy (SURPASS 1), tirzepatide versus semaglutide (SURPASS 2), tirzepatide versus insulin degludec (SURPASS 3), tirzepatide versus glargine (SURPASS 4), and tirzepatide versus insulin glargine (SURPASS 5), with promising results in a significant decrease in HbA1C and reduction in bodyweight by tirezepatide as compared with placebos [112,113,114,115,116]. Furthermore, trials on SURPASS 6 (tirzepatide versus insulin lispro), SURPASS-CVOT (tirzepatide versus dulaglutide), and SURPASS AP-Combo (tirzepatide versus insulin glargine add-on metformin with or without a sulfonylurea) are ongoing. In reference to the issues raised in this review, the SURPASS-CVOT study, in which tirzepatide versus dulaglutide for 3-point MACCE endpoint (CV death, MI and stroke) will be assessed over an estimated maximum of 54 months (estimated study completion in October 2024), arouses our greatest interest. We look forward to finding out the results of this trial.

## 5. Practical Implications and Future Directions

In this review, we have presented the most extensive summary of the role of the most predominant gastrointestinal hormones—GIP and GLP-1—in the pathophysiology of atherosclerosis and CAD both in animals and in humans. We have (1) described GIP and GLP-1 as expressed in many human tissues, (2) emphasized the relationship between GIP and GLP-1 and inflammation, (3) highlighted importance of GIP and GLP-1-dependent pathways in atherosclerosis and CAD, and (4) proved that GIP and GLP-1 could be used as markers of incidence, clinical course, and recurrence of CAD and related to the extent and severity of atherosclerosis and myocardial ischemia. Based on these data and evidence from randomized-controlled trials, three GLP-1 RAs were approved for treatment of DM 2 patients with concomitant CVD or very high/high CV risk and found a place in European Society of Cardiology (ESC) and European Association for the Study of Diabetes (EASD) guidelines from 2019—liraglutide, semaglutide, and dulaglutide [117]. Currently, liraglutide, semaglutide, and dulaglutide are recommended in patients with DM 2 and CVD or at very high/high CV risk to reduce CV events, and liraglutide is recommended in patients with DM 2 and CVD, or at very high/high CV risk, to reduce the risk of death [95]. However, it should be noted that although the association of GLP-1 with atherosclerosis and CAD in animals and humans has been well established, for GIP, this issue has been studied in animal models, but for humans the evidence is poor. Very little about GIP metabolism in the acute phase of MI or for CCS patients is known: the mechanism of direct influence of GIP on cardiomyocytes is poorly understood. More questions remain: (1) Does and it truly work cardioprotectively, and how? (2) Is its effect on CV prognosis dependent or not on the co-occurrence of DM 2. This latter question is especially important remains poorly understood. The translational relevance of body-weight-reducing effects (from rodents to humans) is unclear, and the safety of long-term GIPR agonism and antagonism remains to be uncovered in humans. The question of whether GIPR agonism or antagonism translates into a therapeutic benefit in CV disease requires testing of appropriate pharmacological interventions.

Currently, as we are struggling with the next waves of the coronavirus disease 2019 (COVID-19) pandemic, and thus as limitations arise related to the availability of healthcare, patients with diabetes are doomed to show a worse prognosis because they are exposed to the most severe forms of COVID-19 and related adverse outcomes. DM 2 patients usually present with excess adipose tissue, which enhances chronic inflammatory and prooxidative states. Chronic hyperglycemic and inflammatory states are the two predominant elements of immunosuppression in diabetic patients at higher risk of COVID-19 infection, and also represent an increased risk of mortality. Another problem for patients with DM 2 hospitalized for acute COVID-19 is that, even if they survive, there are adverse aspects of long COVID-19 (or post-COVID-19 syndrome), which affects 10% of those infected and is related to the multi-organ damage caused by acute infection. Long COVID-19 is another factor significantly contributing to the very poor prognosis of diabetic patients. This is why it is so important to deal with the risk factors of a potentially severe course of COVID-19, and new glucose-lowering drugs such as GLP-1 RAs reducing impairment of multi-organ metabolic disorders are the key to improve prognosis and survival of DM 2 patients [118,119].

We believe that due to the widespread and growing interest for the potential use of incretins in CV diseases, further research (randomized-controlled trials) in this direction is desirable. For the future, we would like to recognize GIP and GLP-1 as widely implemented into clinical practice as new biomarkers of ACS and CCS. We hope that GIP and GLP-1 will soon permanently enter the canon of drugs used for CAD-management, also in non-diabetic patients, reducing mortality and the risk of reinfarction or hospitalization for angina pectoris.

## Figures and Tables

**Figure 1 biology-11-00288-f001:**
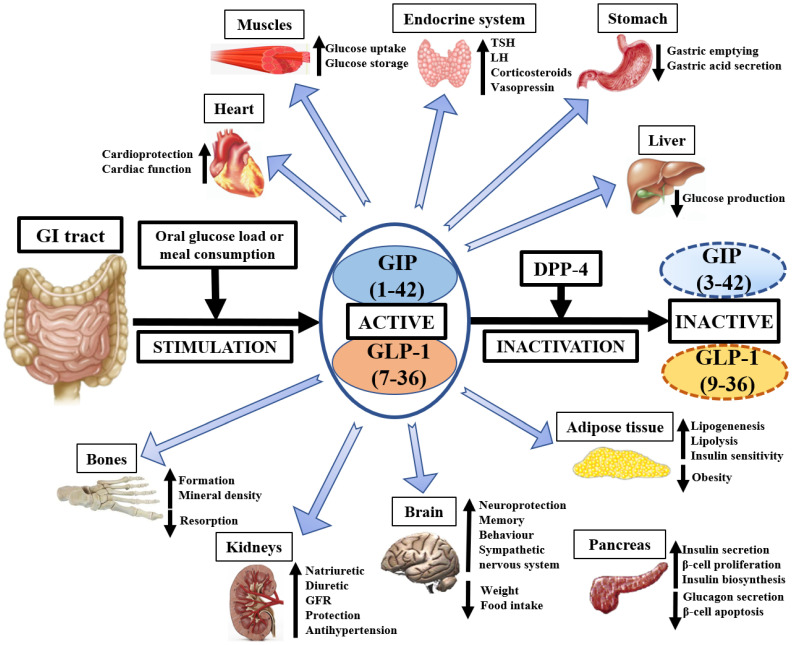
The pleiotropic physiological importance of GIP, GLP-1, and DPP-4 in medicine. GI—gastrointestinal, GIP—glucose-dependent insulinotropic polypeptide, GLP-1—glucagon-like peptide-1, DPP-4—dipeptidyl peptidase-4, GFR—glomerular filtration rate, TSH—thyroid-stimulating hormone, LH—luteinizing hormone.

**Table 1 biology-11-00288-t001:** Animal studies evaluating the role of GIP and GLP-1 in atherosclerosis and myocardial ischemia.

Study Type	Clinical Characteristics	Conclusions	Ref. No.
prospective	6-week-old mice (*n* = 19) with apolipoprotein E knockout (ApoE^−/−^)	Four-week infusion of exendin-4 (GLP-1 RA) reduced monocyte/macrophage accumulation in the arterial wall by inhibiting the inflammatory response in macrophages.	[65]
prospective	17-week-old mice (*n* = 346) with apolipoprotein E knockout (ApoE^−/−^)	Four-week infusion of GLP-1(7–36) or GIP(1–42) significantly suppressed atherosclerotic lesions and macrophage infiltration in the aortic wall.	[66]
prospective	21-week-old mice (*n* = 49) with apolipoprotein E knockout (ApoE^−/−^)	Four-week infusion of GIP significantly suppressed macrophage-driven atherosclerotic lesions and foam cell formation, but this effect was abolished by co-infusions with a GIPR antagonist.	[67]
prospective	17-week-old mice (*n* = 44) with apolipoprotein E knockout (ApoE^−/−^)	GIPR activation decreased atherosclerotic plaque formation and macrophage foam cell formation.	[68]
prospective	6-week-old mice (*n* = 40) with apolipoprotein E knockout (ApoE^−/−^)	GLP-1 overexpression via GLP-1 injection reduced and stabilized atherosclerotic lesions by directly blocking monocyte migration and preventing inflammatory activation of monocytes/macrophages.	[69]
prospective	6-week-old mice (*n* = 18–20) with apolipoprotein E knockout (ApoE^−/−^)	GIP overexpression via GIP injection led to reduced atherosclerotic plaque macrophage infiltration and increased collagen content with no change in overall lesion size, suggesting improved plaque stability. GIP prevented proinflammatory macrophage activation, leading to reduced LPS-induced IL-6 secretion and inhibition of MMP-9 activity.	[70]
prospective	9-week-old mice (*n* = 35)	GIP exerted a protective effect against peripheral arterial remodeling with possible mechanism mediated by NO.	[71]
prospective	107 rats: 22—in vivo; 85—in vitro within 30 min of LM occlusion and 2 h of reperfusion	GLP-1 protects against myocardial infarction and significantly reduced ischemia in the isolated and intact rat heart. This preservation involves multiple prosurvival kinases, such as cAMP, PI3K, and p42/44 mitogen-activated protein kinase.	[72]
prospective	75 rats within 30 min of low-flow ischemia and 30 min of reperfusion	GLP-1 infusion increased myocardial glucose uptake by increasing NO production and enhanced recovery after low-flow ischemia with significant improvements in LVED pressure and LV developed pressure.	[73]
prospective	18 pigs after ischemia by LCx ligation and subsequent reperfusion	Treatment with exenatide (GLP-1 analogue) reduced myocardial infarct size and prevented deterioration of systolic and diastolic cardiac function.	[74]
prospective	Rats (*n* = 22) with MI by LAD ligation	GIP reduced the protein expression levels of resistin (a promoter of cardiac remodeling and dysfunction) and thus may act cardioprotectively.	[75]
prospective	10–12-week-old mice (*n* = 4–6) with MI by LAD ligation	GIP does not impair ventricular function or survival after ischemic cardiac injury. Genetic elimination of the *GIPR (GIPR*^−/−^*)* increased TAG stores and reduced MI-induced ventricular injury and enhanced survival associated with reduced HSL phosphorylation.	[76]
prospective	6-week-old mice (*n* = 12) with MI induced by LAD ligation	Endogenous circulating GLP-1 concentrations were elevated in a murine model of MI. Increased GLP-1 secretion in response to MI was found to be cardioprotective in mice by enhancing LV contractility in a time-dependent manner, independently of glucose metabolism.	[77]

GLP-1 RA—glucagon-like peptide-1 receptor analogue; GIP—glucose-dependent insulinotropic polypeptide; GLP-1—glucagon-like peptide-1; GIPR—glucose-dependent insulinotropic polypeptide receptor; LPS—lipopolysaccharides, endotoxin; IL-6—interleukin-6; MMP-9—matrix metallopeptidase-9; NO—nitric oxide; LM—left main coronary artery; cAMP—cyclic adenosine monophosphate; PI3K—PhosphoInositide 3-Kinase; LVED—left ventricle end-diastolic; LV—left ventricle; LCx—left circumflex coronary artery; LAD—left anterior descending artery; MI—myocardial infarction; TAG—triacylglycerol; HSL—hormone-sensitive lipase.

**Table 2 biology-11-00288-t002:** Human studies involving the role of GIP and GLP-1 in atherosclerosis and coronary artery disease.

Study Type	Clinical Characteristics	Conclusions	Ref. No.
prospective, randomized	69 DM 2 patients (60—completed) treated with metformin plus exenatide (30) or insulin glargine (30)	One-year treatment of exenatide resulted in significant reduction of prandial glucose, triglycerides, and markers of oxidative stress as compared with insulin glargine.	[78]
prospective	64 DM 2 patients with no prior history of CAD	Eight-month treatment of liraglutide with metformin resulted in significant reduction of fasting glucose, HbA1C and the thickness of CIMT, however not affected body weight and waist circumference.	[79]
prospective, randomized	66 DM 2 patients treated with exenatide or insulin aspartate	Fifty-two-week treatment of exenatide more significantly reduced the CIMT thickness, body weight, and lipid markers as compared with insulin aspartate.	[80]
prospective, randomized	28 patients (16 DM 2 and 12 healthy control)	The presence of GLP-1 during hyperglycemia significantly protected endothelial function and decreased hyperglycemia-induced oxidative stress generation.	[81]
retrospective	Patients with confirmed diagnosis of critical limb ischemia (*n* = 85) and healthy controls (*n* = 101).	GIPR activation induced the pro-atherosclerotic factors ET-1 and OPN. Fasting GIP levels were significantly higher in patients with a history of CVD (MI or stroke).	[82]
prospective	21 patients (42.9% diabetes) with MI and LVEF < 40% after successful PCI (GLP-1 = 10, controls = 11).	Seventy-two-hour infusion of GLP-1 significantly improved LVEF, global and regional wall motion, independently of MI location or history of diabetes.	[83]
prospective, randomized	172 patients (6.4% diabetes) undergoing PCI for STEMI (exenatide = 85, controls = 87).	Administration of exenatide (GLP-1 analogue) at the time of reperfusion increased myocardial salvage and reduced infarct size with no difference observed in 3-month LV function or 30-day clinical events.	[84]
prospective, randomized	58 patients (25.9% diabetes) who underwent PCI for STEMI (exenatide = 18, controls = 40).	Application of exenatide (GLP-1 analogue) reduced infarct size, released TnI and CK-MB, and improved LVEF.	[85]
retrospective	731 patients (32.7% diabetes) presented for elective coronary angiography.	Significantly higher concentration of circulating GIP levels in patients with PAD was demonstrated. GIP levels were independently related to PAD after multivariable adjustment for diabetes. No association between GIP and CAD was found.	[70]
prospective, randomized	9340 DM 2 patients with high risk of CV events (liraglutide = 4668, placebo = 4672) from LEADER trial. Median follow-up: 3.8 years	Liraglutide (GLP-1 RA) significantly reduced the incidence of primary endpoint (CV death, nonfatal MI and nonfatal stroke; HR = 0.87), CV death (HR = 0.78), and overall mortality (HR = 0.85). The rates of nonfatal MI, nonfatal stroke, and hospitalization for HF were nonsignificantly lower in the liraglutide group as compared with placebo.	[86]
prospective, randomized	3297 DM 2 patients with high risk of CV events (semaglutide = 1648, placebo = 1649) from SUSTAIN-6 trial. Median follow-up: 2.1 years	Semaglutide (GLP-1 RA) significantly reduced the incidence of primary endpoint (CV death, nonfatal MI and nonfatal stroke; HR = 0.74), expanded composite outcome (CV death, nonfatal MI, nonfatal stroke, revascularization (coronary or peripheral), and hospitalization for UA or HF; HR = 0.74), revascularization (coronary or peripheral) (HR = 0.65) and nonfatal stroke (HR = 0.61), but not overall mortality, CV death, nonfatal MI, and hospitalization for UA or HF.	[87]
prospective, randomized	9901 DM 2 patients with either previous CVD or CV risk (dulaglutide = 4949, placebo = 4952) from REWIND trial. Median follow-up: 5.4 years	Dulaglutide (GLP-1 RA) significantly reduced the risk of primary endpoint (CV death, nonfatal MI and nonfatal stroke; HR = 0.88) and nonfatal stroke (HR = 0.76), but not overall mortality, CV death, nonfatal MI, or hospitalization for UA or HF.	[88]
prospective, randomized	9463 DM 2 patients with CVD (albiglutide = 4731, placebo = 4732) from HARMONY trial. Median follow-up: 1.6 years	Albiglutide (GLP-1 RA) was proved superior to placebo in reduction of primary composite endpoint (CV death, nonfatal MI, and nonfatal stroke; HR = 0.78), expanded composite outcome (CV death, nonfatal MI, nonfatal stroke, and urgent coronary revascularization for UA; HR = 0.78) and fatal or nonfatal MI (HR = 0.75), but not for overall mortality, CV death or stroke (fatal or nonfatal).	[89]
prospective, randomized	3183 DM 2 patients with high CV risk (semaglutide = 1591, placebo = 1592) from PIONEER-6 trial. Median follow-up: 15.9 months	Semaglutide (GLP-1 RA) was noninferior to placebo for primary composite endpoint (CV death, nonfatal MI and nonfatal stroke; HR = 0.79), overall mortality (HR = 0.51) and components of primary outcome: CV death (HR = 0.49), nonfatal MI (HR = 1.18), and nonfatal stroke (HR = 0.74).	[90]
prospective, randomized	14,752 DM 2 patients and with or without CVD (exenatide = 7356, placebo = 7396) form EXSCEL trial. Median follow-up: 3.2 years	Exenatide (GLP-1 RA) was proved noninferior to placebo in reduction of primary composite endpoint (CV death, nonfatal MI, and nonfatal stroke; HR = 0.91). The rates of CV death, fatal or nonfatal MI, fatal or nonfatal stroke, and hospitalization for HF or ACS did not differ significantly between exenatide and placebo.	[91]
prospective, randomized	6068 DM 2 patients with MI or hospitalized for UA within the previous 6 months (lixenatide = 3034, placebo = 3034) from ELIXA trial. Median follow-up: 25 months	Lixenatide (GLP-1 RA) was noninferior to placebo in reduction of primary composite endpoint (CV death, nonfatal MI, nonfatal stroke, or hospitalization for UA; HR = 1.02), individual components of primary composite endpoint, and overall mortality (HR = 1.13).	[92]
prospective, registry	17,868 patients with diabetes discharged alive after a first event of MI (365 (2%) using GLP-1 RAs) from nationwide SWEDEHEART registry. Median follow-up: 3.0 years	Compared to standard of diabetes care, use of GLP-1 RAs was associated with a lower event risk (adjusted HR 0.72), mainly attributed to a lower rate of reinfarction (HR = 0.71) and stroke (HR = 0.42), with no suggestion of heterogeneity across subgroups of age, sex, CKD and STEMI.	[93]
meta-analysis from randomized trials	33,475 DM 2 patients with or without established CVD (but high/very high CV risk). Median follow-up: 2.1–3.8 years	GLP-1 Ras showed a significant reduction in primary composite endpoint (CV death, nonfatal MI, and nonfatal stroke; HR = 0.90), overall mortality (HR = 0.88), and CV death (0.87) with no significant effects observed for fatal and nonfatal MI, fatal and nonfatal stroke. and hospitalization for UA or HF.	[94]
meta-analysis from randomized trials	56,004 DM 2 patients with or without established CVD (but high/very high CV risk). Median follow-up: 1.3–5.4 years	GLP-1 RAs showed a significant reduction in primary composite endpoint (CV death, nonfatal MI and nonfatal stroke; HR = 0.88), overall mortality (HR = 0.89), CV death (HR = 0.88), fatal and nonfatal stroke (HR = 0.84), and hospitalization for HF (HR = 0.92) with no significant effect observed for fatal and nonfatal MI.	[95]
prospective, randomized	41 patients (28 with complete data) with CAD and newly diagnosed DM 2	The combination of liraglutide and metformin reduced total LDL subfractions by reducing the most atherogenic subfraction LDL_5._ The combination of liraglutide and metformin reduced inflammation marker: CRP but not TNF-α.	[96]
prospective	12 patients (10—nondiabetic) presenting with STEMI before and 24, 72 h, and 90 days after PCI.	Mean group GLP-1 levels 24 h after arrival as well as peak levels determined within 72 h were significantly increased as compared with the mean levels before PCI. GLP-1 levels determined 90 days after discharge did not significantly differ as compared to levels at arrival. No correlation was found between GLP-1 levels and any of the clinical and laboratory parameters.	[97]
retrospective	918 patients (75.7%—nondiabetic) with MI (321 STEMI, 597 NSTEMI). Median follow-up: 310 days for primary endpoint and 311 days for all-cause mortality	GLP-1 was found to be significantly associated with combined primary endpoint (CV death, nonfatal MI, and nonfatal stroke) and all-cause mortality (HR of logarithmized GLP-1 values: 6.29 and 5.71, respectively). GLP-1 was found to be a powerful biomarker of CV events and death in patients with MI and a strong indicator of CV risk, especially for early events (30 days from admission), superior to other established biomarkers (hs-TnT, GFR, hs-CRP, NT-proBNP).	[98]
retrospective	41 patients presented with clinical indication for coronary angiography (26-STEMI; 15-controls (angiographic exclusion of CAD)).	Endogenous circulating GLP-1 concentrations was markedly elevated in patients with STEMI, independently of food intake. Activation of GLP-1 system increased left ventricular contractility during MI.	[77]
retrospective	103 patients (78 admitted for PCI) with STEMI (*n* = 33; 20—nondiabetic) and three control groups: NSTEMI (*n* = 27; 14—nondiabetic), stable angina pectoris (*n* = 18; 8—nondiabetic), and control-healthy subjects (*n* = 25).	Admission levels of GLP-1 were significantly increased in patients with STEMI, NSTEMI, and stable angina pectoris as compared with healthy individuals. No correlations between the GLP-1 levels and clinical laboratory and demographic parameters were found.	[99]

DM 2—diabetes mellitus 2; GLP-1—glucagon-like peptide-1; GIPR—glucose-dependent insulinotropic polypeptide receptor; ET-1—endothelin-1; OPN—osteopontin; HbA1C—glycated hemoglobin; CIMT—carotid-intima media thickness; GIP—glucose-dependent insulinotropic polypeptide; CVD—cardiovascular disease; MI—myocardial infarction; LVEF—left ventricle ejection fraction; PCI—percutaneous coronary intervention; STEMI—ST-elevation myocardial infarction; LV—left ventricle; TnI—troponin I; CK-MB—creatine kinase-MB; PAD—peripheral artery disease; CAD—coronary artery disease; CV—cardiovascular; GLP-1 RA—glucagon-like peptide-1 receptor analogue; HR—hazard ratio; HF—heart failure; UA—unstable angina; ACS—acute coronary syndrome; CKD—chronic kidney disease; NSTEMI—non-ST-elevation myocardial infarction; hs-TnT—high-sensitivity troponin T; GFR—glomerular filtration rate; hs-CRP—high-sensitivity C-reactive protein; NT-proBNP—N-terminal prohormone of brain natriuretic peptide.

## Data Availability

Not applicable.

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
