# Peer review of "Gastrointestinal Incretins—Glucose-Dependent Insulinotropic Polypeptide (GIP) and Glucagon-like Peptide-1 (GLP-1) beyond Pleiotropic Physiological Effects Are Involved in Pathophysiology of Atherosclerosis and Coronary Artery Disease—State of the Art"

_biology, 2022, doi:10.3390/biology11020288_

Round 1
Reviewer 1 Report
In this manuscript, Szymon Jonik et al. describe the effects of gastrointestinal incretins on the physiopathology of atherosclerosis and coronary diseases. The review is written in sufficient detail and can make a good scientific contribution in this field. However, there are some comments.
- In the review there is much information about the GLP-1, it would be interesting to discuss more intensely the dual agonists (GIP/GLP-1), which are emerging molecules. In fact in the latest scientific evidence, it has been shown that the co-administration of GLP-1 and GIP, as a single molecule, has higher metabolic and anti-diabetic effects than the administration of individual molecules, in this regard I suggest to see a review published recently (see: doi.org/10.3390/life12010029).
- At the line 250, probably 346 is a typing error.
- About the table 1 and 2, I think it is better to use either the pointed or numbered list in the third column, also each row of the table should have the same style.
Author Response
Dear
We greatly appreciate you having reviewed our manuscript.
Below we presented our responses to your comments.
- Your comment: In the review there is much information about the GLP-1, it would be interesting to discuss more intensely the dual agonists (GIP/GLP-1), which are emerging molecules. In fact in the latest scientific evidence, it has been shown that the co-administration of GLP-1 and GIP, as a single molecule, has higher metabolic and anti-diabetic effects than the administration of individual molecules, in this regard I suggest to see a review published recently (see: doi.org/10.3390/life12010029).
Our answer: We agree that in our article there is much information about GLP-1 and it outweighs the data on GIP. This is because in this article we aimed to include all the data available in the literature on GLP-1 and GIP, and there is simply more data on GLP-1.
We agree that dual agonists (GIP/GLP-1) will be an interesting addition to this review. We have read the review recommended by you with great attention.
We have added new section – page 18, line 639: 4. The dual GIP/GLP-1 agonism.
In page 18-19, line 640-694 we have added:
Very recently, there have been reports in the literature of a dual agonists (GIP/GLP-1) effect as having a positive impact on glycemic control and improving insulin sensitivity, and thus reducing complications related to hyperglycemia. In several animal studies it was demonstrated that co-administration of GLP-1 and GIP improved hypoglycemic, insulinotropic, and total body mass-reducing effects as compared with individual incretins. [100], [101] Moreover, the synergistic action of both GLP-1 and GIP was proved as improving energy consumption and food intake in mice. [100], [102], [103], [104] However, most of these studies did not evaluate the role of glucagon, mainly because its level was not measured. Human studies are providing more data on this topic. The co-infusion of these incretins increased the insulin secretion as compared with single use of GLP-1 or GIP in the study by Nauck MA., et al. [105] Moreover, the incretins co-administration induced a significant glucagonostatic effect, compared to separately infusion of glucose infusion alone. [106] In another study, the insulin level increased after co-infusion as compared with GIP, GLP-1 or saline (placebo) alone in healthy subjects. However, in obese patients with DM 2, the insulin secretion did not rise as compared with GLP-1 infusion alone, but an increase when compared with GIP and placebo administration was detected, which highlighted the reduced insulinotropic effect of GIP in diabetic subjects. [107] In the study by Mentis N, et al., the co-infusion of GLP-1 and GIP as compared with GLP-1 alone resulted in similar blood glucose and insulin secretion rates in diabetic patients, while the suppression of plasma glucagon by GLP-1 was antagonized by GIP. [108] After that, when the co-infusion of GLP-1 and GIP was proved as having favorable effect on glucose and bodyweight control, the studies concerning dual molecule GLP-1R/GIPR as twincretin have been started. In 2013, Finan B, et al. demonstrated that unimolecular dual incretin administrated to rodent models of obesity and diabetes improved insulin sensitivity and increased pancreatic insulin deficiency, corrected hyperglycemia and adiposity-induced insulin resistance and lowered body weight. [109] The beneficial effect of GLP-1R/GIPR dual agonist was further confirmed in clinical trials. The double-blind, placebo-controlled trial with a fatty-acylated GIP/GLP-1 dual agonist (NNC0090-2746, Phase 1 and Phase 2) was performed in patients with DM-2 inadequately controlled with metformin administered with 1.8 mg of NNC 0090-2746 as compared with 1.8 mg of liraglutide. After two weeks of treatment, NNC0090-2746 decreased HbA1c, fasting, and postprandial plasma glucose. NNC0090-2746 reduced HbA1c similarly to the group treated with liraglutide, whereas the bodyweight reduction was significantly greater than that of liraglutide. The treatment with NNC0090-2746 was generally safe and well tolerated. [110] Furthermore, an another GLP-1R/GIPR dual agonist, LY3298176 was also tested (Phase 2). This double-blind, randomized study was performed in DM-2 subjects, received either once weekly subcutaneous LY3298176 (1 mg, 5 mg, 10 mg, or 15 mg), dulaglutide (1,5 mg), or placebo. After 26-weeks of treatment LY3298176 significantly reduced HbA1c in a dose-dependent manner and reduced body weight as compared with dulaglutide or placebo with similar to dulaglutide adverse effects and no reports of severe hypoglycemia. [111] Based on promising results from Phase 1 and 2 clinical trials, a Phase 3 clinical trial of the dual GIP/GLP-1 RA (LY3298176) named tirzepatide was initiated. To date, the SURPASS program includes several ongoing randomized controlled trials for DM-2 patients: tirzepatide in monotherapy (SURPASS 1), tirzepatide versus semaglutide (SURPASS 2), tirzepatide versus insulin degludec (SURPASS 3), tirzepatide versus glargine (SURPASS 4) and tirzepatide versus insulin glargine (SURPASS 5) with promising results in significant decrease of HbA1C and reduction of bodyweight by tirezepatide as compared with placebos. [112], [113], [114], [115], [116] Furthermore, SURPASS 6 (tirzepatide versus insulin lispro), SURPASS-CVOT (tirzepatide versus dulaglutide) and SURPASS AP-Combo (tirzepatide versus insulin glargine add-on metformin with or without a sulfonylurea) are ongoing. As referring to the issues raised in this review, the SURPASS-CVOT study, in which tirzepatide versus dulaglutide for 3-point MACCE endpoint (CV death, MI and stroke) will be assessed for over an estimated maximum of 54 months (estimated study completion in October 2024). We look forward to find out the results of this trial.
And in Section References – page 25-26, line 1033-1089 we have added 17 new references corresponding to this section.
Gault VA, Kerr BD, Harriott P, Flatt PR. Administration of an acylated GLP-1 and GIP preparation provides added beneficial glucose-lowering and insulinotropic actions over single incretins in mice with Type2 diabetes and obesity. Clin. Sci. 2011; 121: 107–117. doi: 10.1042/CS20110006.- Bergmann NC, Lund A, Gasbjerg LS, Meessen E, Andersen MM, Bergmann S, Hartmann B, Holst JJ, Jessen L, Christensen MB, Vilsbøll T, Knop FK. Effects of combined GIP and GLP-1 infusion on energy intake, appetite and energy expenditure in overweight/obese individuals: A randomised, crossover study. Diabetologia 2019; 62: 665–675. doi: 10.1007/s00125-018-4810-0.
- Irwin N, McClean PL, Cassidy RS, O’harte FP, Green BD, Gault VA, Harriott P, Flatt PR. Comparison of the antidiabetic effects of GIP- and GLP-1-receptor activation in obese diabetic (ob/ob) mice: Studies with DPP IV resistant N-AcGIP and exendin(1-39)amide. DiabetesMetab. Res. Rev. 2007; 23: 572–579. doi: 10.1002/dmrr.729.
- Irwin N, Hunter K, Frizzell N, Flatt PR. Antidiabetic effects of sub-chronic activation of the GIP receptor alone and in combination with background exendin-4 therapy in high fat fed mice. Regul. Pept. 2009; 153: 70–76. doi: 10.1016/j.regpep.2008.11.007.
- Irwin N, McClean PL, Flatt PR. Comparison of the subchronic antidiabetic effects of DPP IV-resistant GIP and GLP-1 analogues in obese diabetic (ob/ob) mice. J Pept Sci. 2007; 13: 400–405. doi: 10.1002/psc.861.
- Nauck MA, Bartels E, Orskov C, Ebert R, Creutzfeldt W. Additive insulinotropic effects of exogenous synthetic human gastric inhibitory polypeptide and glucagon-like peptide-1-(7-36) amide infused at near-physiological insulinotropic hormone and glucose concentrations. J Clin Endocrinol Metab. 1993; 76: 912–917. doi: 10.1210/jcem.76.4.8473405.
- Elahi D, McAloon-Dyke M, Fukagawa NK, Meneilly GS, Sclater AL, Minaker KL, Habener JF, Andersen DK. The insulinotropic actions of glucose-dependent insulinotropic polypeptide (GIP) and glucagon-like peptide-1 (7-37) in normal and diabetic subjects. Regul Pept. 1994; 51: 63–74. doi: 10.1016/0167-0115(94)90136-8.
- Daousi C, Wilding JP, Aditya S, Durham BH, Cleator J, Pinkney JH, Ranganath LR. Effects of peripheral administration of synthetic human glucose-dependent insulinotropic peptide (GIP) on energy expenditure and subjective appetite sensations in healthy normal weight subjects and obese patients with type 2 diabetes. Clin Endocrinol. 2009; 71: 195–201. doi: 10.1111/j.1365-2265.2008.03451.x.
- Mentis N, Vardarli I, Köthe LD, Holst JJ, Deacon CF, Theodorakis M, Meier JJ, Nauck MA. GIP does not potentiate the antidiabetic effects of GLP-1 in hyperglycemic patients with type 2 diabetes. Diabetes 2011; 60: 1270–1276. doi: 10.2337/db10-1332.
109. Finan B, Ma T, Ottaway N, Müller TD, Habegger KM, Heppner KM, Kirchner H, Holland J, Hembree J, Raver C, Lockie SH, Smiley DR, Gelfanov V, Yang B, Hofmann S, Bruemmer D, Drucker DJ, Pfluger PT, Perez-Tilve D, Gidda J, Vignati L, Zhang L, Hauptman JB, Lau M, Brecheisen M, Uhles S, Riboulet W, Hainaut E, Sebokova E, Conde-Knape K, Konkar A, DiMarchi RD, Tschöp MH. Unimolecular dual incretins maximize metabolic benefits in rodents, monkeys, and humans. Sci Transl Med. 2013; 5: 209ra151. doi: 10.1126/scitranslmed.3007218. - Schmitt C, Portron A, Jadidi S, Sarkar N, DiMarchi R. Pharmacodynamics, pharmacokinetics and safety of multiple ascending doses of the novel dual glucose-dependent insulinotropic polypeptide/glucagon-like peptide-1 agonist RG7697 in people with type 2 diabetes mellitus. Diabetes Obes Metab. 2017; 19: 1436–1445. doi: 10.1111/dom.13024.
- Frias JP, Nauck MA, Van J, Kutner ME, Cui X, Benson C, Urva S, Gimeno RE, Milicevic Z, Robins, D, Haupt A. Efficacy and safety of LY3298176, a novel dual GIP and GLP-1 receptor agonist, in patients with type 2 diabetes: A randomised, placebo-controlled and active comparator-controlled phase 2 trial. Lancet 2018; 392: 2180–2193. doi: 10.1016/S0140-6736(18)32260-8.
- Rosenstock J, Wysham C, Frías JP, Kaneko S, Lee CJ, Fernández Landó L, Mao H, Cui X, Karanikas CA, Thieu VT. Efficacy and safety of a novel dual GIP and GLP-1 receptor agonist tirzepatide in patients with type 2 diabetes (SURPASS-1): A double-blind, randomised, phase 3 trial. Lancet 2021; 398: 143–155. doi: 10.1016/S0140-6736(21)01324-6.
- Frías JP, Davies MJ, Rosenstock J, Pérez Manghi FC, Fernández Landó L, Bergman BK, Liu B, Cui X, Brown K, SURPASS-2 Investigators. Tirzepatide versus Semaglutide Once Weekly in Patients with Type 2 Diabetes. N Engl J Med. 2021; 385: 503–515. doi: 10.1056/NEJMoa2107519.
- Ludvik B, Giorgino F, Jódar E, Frias JP, Fernández Landó L, Brown K, Bray R, Rodríguez Á. Once-weekly tirzepatide versus once-daily insulin degludec as add-on to metformin with or without SGLT2 inhibitors in patients with type 2 diabetes (SURPASS-3): A randomised, open-label, parallel-group, phase 3 trial. Lancet 2021; 398: 583–598. doi: 10.1016/S0140-6736(21)01443-4.
- Del Prato S, Kahn SE, Pavo I, Weerakkody GJ, Yang Z, Doupis J, Aizenberg D, Wynne AG, Riesmeyer JS, Heine RJ, Wiese RJ, SURPASS-4 Investigators. Tirzepatide versus insulin glargine in type 2 diabetes and increased cardiovascular risk (SURPASS-4): A randomised, open-label, parallel-group, multicentre, phase 3 trial. Lancet 2021;398:811–824. doi: 10.1016/S0140-6736(21)02188-7.
- A Study of Tirzepatide (LY3298176) Versus Placebo in Participants with Type 2 Diabetes Inadequately Controlled on Insulin Glargine With or Without Metformin. Available online: https://clinicaltrials.gov/ct2/show/NCT04039503.
- Your comment: At the line 250, probably 346 is a typing error.
Our answer: No, this is not error – this is the number of mice used in the study by Nagashima, et al. We have corrected it to – page 7, line 257-258: “A total number of 346….male mice.”
3. Your comment: About the table 1 and 2, I think it is better to use either the pointed or numbered list in the third column, also each row of the table should have the same style.
Our answer: Yes, you are right. We used initially only pointed list in the third column, but it was changed by journal converter during online submission, and hence this slight inaccuracy. We have corrected it. We have also corrected style of each row of the table. This new style is ok for you
Reviewer 2 Report
It’s a very interesting and well-written review article, which needs some improvement.
Section 1. Add a brief comment about the recent data from the CAPTURE study, showing that the vast majority of T2DM has an atherosclerotic form of cardiovascular disease (Cardiovasc Diabetol 2021;20:154) and that GLP-1 RAs have a direct anti-atherosclerotic effect.
Section 3.1. Add clinical findings with the use of GLP-1 RA liraglutide in T2DM patients showing a reduction in atherogenic small dense LDL (Atherosclerosis. 2019;288:60–6). This effects has been proposed as an important novel anti-atherogenic effect due to the well known association between small dense LDL and cardiovascular diseases.
Table 2. Add two studies with the use of exenatide and liraglutide showing a reduction in subclinical atherosclerosis. Also, include the first clinical study performed on T2DM that showed a reduction in oxidative stress with the use of a GLP-1 RA. All these studies should be included in Table 2 after the first study included in this table, which is the study by Ceriello et al. on endothelial dysfunction.
Section 4. Please mention that GLP-1 RAs have a particular importance during current pandemic to reduce diabetic cardiometabolic risk, since diabetic patients are those exposed to the most severe forms of COVID and related mortality: insights from recent experience can guide future management of diabetic patients in general (Metab Syndr Relat Disord 2020;18:173-175) and also of those with long-COVID (Nat Rev Endocrinol 2021;17:379-380).
Author Response
Dear
We greatly appreciate you having reviewed our manuscript.
Below we presented our responses to your comments.
- Your comment: Section 1. Add a brief comment about the recent data from the CAPTURE study, showing that the vast majority of T2DM has an atherosclerotic form of cardiovascular disease (Cardiovasc Diabetol 2021;20:154) and that GLP-1 RAs have a direct anti-atherosclerotic effect.
Our answer – page 6, line 230-236 – we have added: The real-life evidences demonstrating that the vast majority of DM2 has an atherosclerotic form of cardiovascular (CV) disease are derived from recently published results of CAPTURE study. In this multicenter trial assessing worldwide prevalence of CV disease in nearly 10,000 people with DM2, CV disease was diagnosed in one in three patients with DM2, whereas overall prevalence rates of weighted and atherosclerotic CV disease were 34.8% and 31.8%, respectively. Furthermore, in this trial, GLP-1RA or SGLT-2 inhibitors were proved as playing beneficial CV effects even in patients with established CV disease. [64]
and we have added corresponding reference to this issue – section References, page 23, line 908-911:
- Mosenzon O, Alguwaihes A, Leon JLA, Bayram F, Darmon P, Davis TME, Dieuzeide G, Eriksen KT, Hong T, Kaltoft MS, Lengyel C, Rhee NA, Russo GT, Shirabe S, Urbancova K, Vencio S, CAPTURE Study Investigators. CAPTURE: a multinational, cross-sectional study of cardiovascular disease prevalence in adults with type 2 diabetes across 13 countries. Cardiovasc Diabetol. 2021; 20(1): 154. doi: 10.1186/s12933-021-01344-0.
- Your comment: Section 3.1. Add clinical findings with the use of GLP-1 RA liraglutide in T2DM patients showing a reduction in atherogenic small dense LDL (Atherosclerosis. 2019;288:60–6). This effects has been proposed as an important novel anti-atherogenic effect due to the well known association between small dense LDL and cardiovascular diseases.
Our answer – page 14, line 560-569 – we have added: In another study Anholm C, et al. investigated the effect of liraglutide combined with metformin on lipid particle sub-fractions in 41 patients (28 with complete follow-up) with newly diagnosed DM2 and at high risk of CV events, i.e. with CAD. Overall, liraglutide did not affect lipid subfractions or markers of low-grade inflammation (LGI) compared to placebo. However, the combination of liraglutide and metformin significantly reduced the most atherogenic subfraction – small, dense LDL5 particles despite the population being well-controlled on stable statin therapy. Thus, the investigators proved that antidiabetic drugs using for DM2-patients with CAD may reduce CV risk and improve lipid profile by lowering the level of small dense LDL. [96]
and we have added:
1) summary of this issue in Table 2 – page 18:
2) corresponding reference to this issue – section References, page 25, line 1020-1023:
- Anholm C, Kumarathurai P, Pedersen LR, Samkani A, Walzem RL, Nielsen OW, Kristiansen OP, Fenger M, Madsbad S, Sajadieh A, Haaugard SB. Liraglutide in combination with metformin may improve the atherogenic lipid profile and decrease C-reactive protein level in statin treated obese patients with coronary artery disease and newly diagnosed type 2 diabetes: A randomized trial. Atherosclerosis 2019; 288: 60-66. doi: 10.1016/j.atherosclerosis.2019.07.007.
- Your comment: Table 2. Add two studies with the use of exenatide and liraglutide showing a reduction in subclinical atherosclerosis. Also, include the first clinical study performed on T2DM that showed a reduction in oxidative stress with the use of a GLP-1 RA. All these studies should be included in Table 2 after the first study included in this table, which is the study by Ceriello et al. on endothelial dysfunction.
Our answer – we have added all articles you had requested:
We have added – page 12, line 425-452: In the first randomized-controlled study investigating association between GLP-1 RA and oxidative stress by Bunck MC, et al. 60 DM-2 with metformin-treated patients were randomized to receive exenatide or glargine insulin (ratio 1:1) for one year. A 51-week of exenatide treatment resulted in significant reduction of plasma concentration of oxidative stress markers: malondialdehyde – MDA (compared to baseline and glargine insulin; P<0.001) and oxidized low-density lipoprotein – oxLDL (compared to baseline; P<0.05). Moreover, the investigators reported significant decrease in postprandial glucose and triglycerides in exenatide-cohort as compared to baseline and glargine insulin. (P<0.01) Changes in markers of oxidative stress were independent of treatment arm and related to changes in postprandial glucose and triglyceride excursions. [78]
In 2014 Rizzo M, et al. presented the effect of GLP-1 RA – liraglutide on subclinical atherosclerosis. In this prospective study, 64 DM2-patients with no prior history of CAD were evaluated. The investigators aimed to answer if adding liraglutide to metformin affects CIMT. After 8 months with liraglutide treatment CIMT significantly decreased from 1.19 to 0.94 (P = 0.001), while no significant reduction in body weight and waist circumference were observed. However, adding liraglutide to metformin resulted in significant decrease in fasting glucose (-2.1 mmol/L, P<0.01) and HbA1C (-1.9 %, P<0.01). This results should be, however considered carefully as in the present study the placebo group was absent. [79] Furthermore, Zhang J, et al. in the first randomized-controlled trial demonstrated the beneficial effect of GLP-1 RA on subclinical atherosclerosis via preventing atherosclerotic progression, as assessed by carotid-intima media thickness (CIMT). A total number of 66 DM2-patients were randomized to receive twice-daily exenatide or aspartate insulin for 52 weeks. Exenatide was proved to reduce the CIMT from baseline more significantly as compared with insulin after 52 weeks with a mean difference of − 0.14 mm (P=0.016). The components of an explanatory endpoint (differences in body weight and lipid metabolism markers) were also more significantly reduced by exenatide at weeks 16 and 40 as compared with insulin. The correlation analyses showed that CIMT was positively associated with low-density lipoprotein cholesterol. [80]
and we have added:
1) summary of this issue in Table 2 – page 16:
2) corresponding reference to this issue – section References, page 24, line 952-960:
- Bunck MC, Cornér A, Eliasson B, Heine RJ, Shaginian RM, Wu Y, Yan P, Smith U, Yki-Järvinen H, Diamant M, Taskinen MR. One-year treatment with exenatide vs. insulin glargine: effects on postprandial glycemia, lipid profiles, and oxidative stress. Atherosclerosis 2010; 212(1): 223-229. doi: 10.1016/j.atherosclerosis.2010.04.024.
- Rizzo M, Chandalia M, Patti AM, Di Bartolo V, Rizvi AA, Montalto G, Abate N. Liraglutide decreases carotid intima-media thickness in patients with type 2 diabetes: 8-month prospective pilot study. Cardiovasc Diabetol. 2014; 13: 49. doi: 10.1186/1475-2840-13-49.
- Zhang J, Xian TZ, Wu MX, Li C, Pan Q, Guo LX. Comparison of the effects of twice-daily exenatide and insulin on carotid intima-media thickness in type 2 diabetes mellitus patients: a 52-week randomized, open-label, controlled trial. Cardiovasc Diabetol. 2020; 19: 48. doi: 10.1186/s12933-020-01014-7.
- Your comment: Section 4. Please mention that GLP-1 RAs have a particular importance during current pandemic to reduce diabetic cardiometabolic risk, since diabetic patients are those exposed to the most severe forms of COVID and related mortality: insights from recent experience can guide future management of diabetic patients in general (Metab Syndr Relat Disord 2020;18:173-175) and also of those with long-COVID (Nat Rev Endocrinol 2021;17:379-380).
Our answer - we have added all articles you had requested:
We have added – page 20, line 723-738: Currently, when we are struggling with the next waves of the coronavirus disease 2019 (COVID-19) pandemic and thus limitations related to the availability of healthcare, patients with diabetes are doomed to a worse prognosis, because they are exposed to the most severe forms of COVID-19 and related adverse outcomes. DM2 – patients usually present with excess adipose tissue, which enhances chronic inflammatory and prooxidative states. Chronic hyperglycemic and inflammatory states are the two predominant elements of immunosuppression in diabetic patients at higher risk of COVID-19 infection, and also represent an increased risk of mortality. Another problem for patients with DM2 hospitalized for acute COVID-19, even if they survive, is the adverse aspect of long COVID (or post-COVID syndrome), which affects 10 % of those infected and is related to the multi-organ damage caused by acute infection. The long COVID is the another factor significantly contributing to the very poor prognosis of diabetic patients. That is why it is so important to deal with the risk factors of a potentially severe course of COVID-19, and new glucose-lowering drugs as GLP-1 RAs due reducing impairment of multi-organ metabolic disorders are the key to improve prognosis and survival of DM-2 patients. [118], [119]
and we have added corresponding reference to this issue – section References, page 26, line 1095-1098:
- Stoian AP, Banerjee Y, Rizvi AA, Rizo M. Diabetes and the COVID-19 Pandemic: How Insights from Recent Experience Might Guide Future Management. Metab Syndr Relat Disord. 2020; 18(4): 173-175. doi: 10.1089/met.2020.0037.
- Khunti K, Davies MJ, Kosiborod MN, Nauck MA. Long COVID - metabolic risk factors and novel therapeutic management. Nat Rev Endocrinol 2021; 17(7): 379-380. doi: 10.1038/s41574-021-00495-0.
Round 2
Reviewer 2 Report
accept in present form
Author Response
Dear Reviewers
We greatly appreciate you having reviewed our manuscript.
Below we presented our responses to your comments.
Reviewer 1
Dear
We have already answered all your inquires.
Below we have presented our current version.
- Your comment: In the review there is much information about the GLP-1, it would be interesting to discuss more intensely the dual agonists (GIP/GLP-1), which are emerging molecules. In fact in the latest scientific evidence, it has been shown that the co-administration of GLP-1 and GIP, as a single molecule, has higher metabolic and anti-diabetic effects than the administration of individual molecules, in this regard I suggest to see a review published recently (see: doi.org/10.3390/life12010029).
Our answer: We agree that in our article there is much information about GLP-1 and it outweighs the data on GIP. This is because in this article we aimed to include all the data available in the literature on GLP-1 and GIP, and there is simply more data on GLP-1.
We agree that dual agonists (GIP/GLP-1) will be an interesting addition to this review. We have read the review recommended by you with great attention.
We have added new section – page 18, line 639: 4. The dual GIP/GLP-1 agonism.
In page 18-19, line 640-694 we have added:
Very recently, there have been reports in the literature of a dual agonists (GIP/GLP-1) effect as having a positive impact on glycemic control and improving insulin sensitivity, and thus reducing complications related to hyperglycemia. In several animal studies it was demonstrated that co-administration of GLP-1 and GIP improved hypoglycemic, insulinotropic, and total body mass-reducing effects as compared with individual incretins. [100], [101] Moreover, the synergistic action of both GLP-1 and GIP was proved as improving energy consumption and food intake in mice. [100], [102], [103], [104] However, most of these studies did not evaluate the role of glucagon, mainly because its level was not measured. Human studies are providing more data on this topic. The co-infusion of these incretins increased the insulin secretion as compared with single use of GLP-1 or GIP in the study by Nauck MA., et al. [105] Moreover, the incretins co-administration induced a significant glucagonostatic effect, compared to separately infusion of glucose infusion alone. [106] In another study, the insulin level increased after co-infusion as compared with GIP, GLP-1 or saline (placebo) alone in healthy subjects. However, in obese patients with DM 2, the insulin secretion did not rise as compared with GLP-1 infusion alone, but an increase when compared with GIP and placebo administration was detected, which highlighted the reduced insulinotropic effect of GIP in diabetic subjects. [107] In the study by Mentis N, et al., the co-infusion of GLP-1 and GIP as compared with GLP-1 alone resulted in similar blood glucose and insulin secretion rates in diabetic patients, while the suppression of plasma glucagon by GLP-1 was antagonized by GIP. [108] After that, when the co-infusion of GLP-1 and GIP was proved as having favorable effect on glucose and bodyweight control, the studies concerning dual molecule GLP-1R/GIPR as twincretin have been started. In 2013, Finan B, et al. demonstrated that unimolecular dual incretin administrated to rodent models of obesity and diabetes improved insulin sensitivity and increased pancreatic insulin deficiency, corrected hyperglycemia and adiposity-induced insulin resistance and lowered body weight. [109] The beneficial effect of GLP-1R/GIPR dual agonist was further confirmed in clinical trials. The double-blind, placebo-controlled trial with a fatty-acylated GIP/GLP-1 dual agonist (NNC0090-2746, Phase 1 and Phase 2) was performed in patients with DM-2 inadequately controlled with metformin administered with 1.8 mg of NNC 0090-2746 as compared with 1.8 mg of liraglutide. After two weeks of treatment, NNC0090-2746 decreased HbA1c, fasting, and postprandial plasma glucose. NNC0090-2746 reduced HbA1c similarly to the group treated with liraglutide, whereas the bodyweight reduction was significantly greater than that of liraglutide. The treatment with NNC0090-2746 was generally safe and well tolerated. [110] Furthermore, an another GLP-1R/GIPR dual agonist, LY3298176 was also tested (Phase 2). This double-blind, randomized study was performed in DM-2 subjects, received either once weekly subcutaneous LY3298176 (1 mg, 5 mg, 10 mg, or 15 mg), dulaglutide (1,5 mg), or placebo. After 26-weeks of treatment LY3298176 significantly reduced HbA1c in a dose-dependent manner and reduced body weight as compared with dulaglutide or placebo with similar to dulaglutide adverse effects and no reports of severe hypoglycemia. [111] Based on promising results from Phase 1 and 2 clinical trials, a Phase 3 clinical trial of the dual GIP/GLP-1 RA (LY3298176) named tirzepatide was initiated. To date, the SURPASS program includes several ongoing randomized controlled trials for DM-2 patients: tirzepatide in monotherapy (SURPASS 1), tirzepatide versus semaglutide (SURPASS 2), tirzepatide versus insulin degludec (SURPASS 3), tirzepatide versus glargine (SURPASS 4) and tirzepatide versus insulin glargine (SURPASS 5) with promising results in significant decrease of HbA1C and reduction of bodyweight by tirezepatide as compared with placebos. [112], [113], [114], [115], [116] Furthermore, SURPASS 6 (tirzepatide versus insulin lispro), SURPASS-CVOT (tirzepatide versus dulaglutide) and SURPASS AP-Combo (tirzepatide versus insulin glargine add-on metformin with or without a sulfonylurea) are ongoing. As referring to the issues raised in this review, the SURPASS-CVOT study, in which tirzepatide versus dulaglutide for 3-point MACCE endpoint (CV death, MI and stroke) will be assessed for over an estimated maximum of 54 months (estimated study completion in October 2024). We look forward to find out the results of this trial.
And in Section References – page 25-26, line 1033-1089 we have added 17 new references corresponding to this section.
- Gault VA, Kerr BD, Harriott P, Flatt PR. Administration of an acylated GLP-1 and GIP preparation provides added beneficial glucose-lowering and insulinotropic actions over single incretins in mice with Type2 diabetes and obesity. Clin. Sci. 2011; 121: 107–117. doi: 10.1042/CS20110006.
- Bergmann NC, Lund A, Gasbjerg LS, Meessen E, Andersen MM, Bergmann S, Hartmann B, Holst JJ, Jessen L, Christensen MB, Vilsbøll T, Knop FK. Effects of combined GIP and GLP-1 infusion on energy intake, appetite and energy expenditure in overweight/obese individuals: A randomised, crossover study. Diabetologia 2019; 62: 665–675. doi: 10.1007/s00125-018-4810-0.
- Irwin N, McClean PL, Cassidy RS, O’harte FP, Green BD, Gault VA, Harriott P, Flatt PR. Comparison of the antidiabetic effects of GIP- and GLP-1-receptor activation in obese diabetic (ob/ob) mice: Studies with DPP IV resistant N-AcGIP and exendin(1-39)amide. DiabetesMetab. Res. Rev. 2007; 23: 572–579. doi: 10.1002/dmrr.729.
- Irwin N, Hunter K, Frizzell N, Flatt PR. Antidiabetic effects of sub-chronic activation of the GIP receptor alone and in combination with background exendin-4 therapy in high fat fed mice. Regul. Pept. 2009; 153: 70–76. doi: 10.1016/j.regpep.2008.11.007.
- Irwin N, McClean PL, Flatt PR. Comparison of the subchronic antidiabetic effects of DPP IV-resistant GIP and GLP-1 analogues in obese diabetic (ob/ob) mice. J Pept Sci. 2007; 13: 400–405. doi: 10.1002/psc.861.
- Nauck MA, Bartels E, Orskov C, Ebert R, Creutzfeldt W. Additive insulinotropic effects of exogenous synthetic human gastric inhibitory polypeptide and glucagon-like peptide-1-(7-36) amide infused at near-physiological insulinotropic hormone and glucose concentrations. J Clin Endocrinol Metab. 1993; 76: 912–917. doi: 10.1210/jcem.76.4.8473405.
- Elahi D, McAloon-Dyke M, Fukagawa NK, Meneilly GS, Sclater AL, Minaker KL, Habener JF, Andersen DK. The insulinotropic actions of glucose-dependent insulinotropic polypeptide (GIP) and glucagon-like peptide-1 (7-37) in normal and diabetic subjects. Regul Pept. 1994; 51: 63–74. doi: 10.1016/0167-0115(94)90136-8.
- Daousi C, Wilding JP, Aditya S, Durham BH, Cleator J, Pinkney JH, Ranganath LR. Effects of peripheral administration of synthetic human glucose-dependent insulinotropic peptide (GIP) on energy expenditure and subjective appetite sensations in healthy normal weight subjects and obese patients with type 2 diabetes. Clin Endocrinol. 2009; 71: 195–201. doi: 10.1111/j.1365-2265.2008.03451.x.
- Mentis N, Vardarli I, Köthe LD, Holst JJ, Deacon CF, Theodorakis M, Meier JJ, Nauck MA. GIP does not potentiate the antidiabetic effects of GLP-1 in hyperglycemic patients with type 2 diabetes. Diabetes 2011; 60: 1270–1276. doi: 10.2337/db10-1332.
- Finan B, Ma T, Ottaway N, Müller TD, Habegger KM, Heppner KM, Kirchner H, Holland J, Hembree J, Raver C, Lockie SH, Smiley DR, Gelfanov V, Yang B, Hofmann S, Bruemmer D, Drucker DJ, Pfluger PT, Perez-Tilve D, Gidda J, Vignati L, Zhang L, Hauptman JB, Lau M, Brecheisen M, Uhles S, Riboulet W, Hainaut E, Sebokova E, Conde-Knape K, Konkar A, DiMarchi RD, Tschöp MH. Unimolecular dual incretins maximize metabolic benefits in rodents, monkeys, and humans. Sci Transl Med. 2013; 5: 209ra151. doi: 10.1126/scitranslmed.3007218.
- Schmitt C, Portron A, Jadidi S, Sarkar N, DiMarchi R. Pharmacodynamics, pharmacokinetics and safety of multiple ascending doses of the novel dual glucose-dependent insulinotropic polypeptide/glucagon-like peptide-1 agonist RG7697 in people with type 2 diabetes mellitus. Diabetes Obes Metab. 2017; 19: 1436–1445. doi: 10.1111/dom.13024.
- Frias JP, Nauck MA, Van J, Kutner ME, Cui X, Benson C, Urva S, Gimeno RE, Milicevic Z, Robins, D, Haupt A. Efficacy and safety of LY3298176, a novel dual GIP and GLP-1 receptor agonist, in patients with type 2 diabetes: A randomised, placebo-controlled and active comparator-controlled phase 2 trial. Lancet 2018; 392: 2180–2193. doi: 10.1016/S0140-6736(18)32260-8.
- Rosenstock J, Wysham C, Frías JP, Kaneko S, Lee CJ, Fernández Landó L, Mao H, Cui X, Karanikas CA, Thieu VT. Efficacy and safety of a novel dual GIP and GLP-1 receptor agonist tirzepatide in patients with type 2 diabetes (SURPASS-1): A double-blind, randomised, phase 3 trial. Lancet 2021; 398: 143–155. doi: 10.1016/S0140-6736(21)01324-6.
- Frías JP, Davies MJ, Rosenstock J, Pérez Manghi FC, Fernández Landó L, Bergman BK, Liu B, Cui X, Brown K, SURPASS-2 Investigators. Tirzepatide versus Semaglutide Once Weekly in Patients with Type 2 Diabetes. N Engl J Med. 2021; 385: 503–515. doi: 10.1056/NEJMoa2107519.
- Ludvik B, Giorgino F, Jódar E, Frias JP, Fernández Landó L, Brown K, Bray R, Rodríguez Á. Once-weekly tirzepatide versus once-daily insulin degludec as add-on to metformin with or without SGLT2 inhibitors in patients with type 2 diabetes (SURPASS-3): A randomised, open-label, parallel-group, phase 3 trial. Lancet 2021; 398: 583–598. doi: 10.1016/S0140-6736(21)01443-4.
- Del Prato S, Kahn SE, Pavo I, Weerakkody GJ, Yang Z, Doupis J, Aizenberg D, Wynne AG, Riesmeyer JS, Heine RJ, Wiese RJ, SURPASS-4 Investigators. Tirzepatide versus insulin glargine in type 2 diabetes and increased cardiovascular risk (SURPASS-4): A randomised, open-label, parallel-group, multicentre, phase 3 trial. Lancet 2021;398:811–824. doi: 10.1016/S0140-6736(21)02188-7.
- A Study of Tirzepatide (LY3298176) Versus Placebo in Participants with Type 2 Diabetes Inadequately Controlled on Insulin Glargine With or Without Metformin. Available online: https://clinicaltrials.gov/ct2/show/NCT04039503.
In the request 1: online resubmission creator changed numbers of references omitting reference no. 109 and changing the successive reference numbers from 109 to 115. We have changed numbers of these references into 110 to 116. - Your comment: At the line 250, probably 346 is a typing error.
Our answer: No, this is not error – this is the number of mice used in the study by Nagashima, et al. Currently, in page 7, line 257-258, we have corrected it to: “A total number of 346….male mice.”
Here too there was an error in the numbering – we have corrected it.
- Your comment: About the table 1 and 2, I think it is better to use either the pointed or numbered list in the third column, also each row of the table should have the same style.
Our answer: Yes, you are right. We used initially only pointed list in the third column, but it was changed by journal converter during online submission, and hence this slight inaccuracy. We have corrected it. We have also corrected style of each row of the table. This new style is ok for you ?
We have corrected style in column no. third and unified style of each rows. Now is it ok for you ?
Reviewer 2
Dear
Thank you for your reviewing.
